# All-optical voltage interrogation for probing synaptic plasticity in vivo

Jacques Carolan[1] ✉, Michelle A. Land [2], Xiaoyu Lu[2,3,8], Maxime Beau[1], Dimitar Kostadinov[1,4], François St-Pierre[2,3,5,6], Beverley A. Clark[1] & Michael Häusser [1,7] ✉

Measuring synaptic efficacy and defining the rules for induction of synaptic plasticity at identified connections in the mammalian brain is essential for understanding how synapses contribute to learning and memory. This requires new approaches to selectively evoke presynaptic activity and measure postsynaptic responses with high spatiotemporal resolution and high sensitivity over long periods in vivo. Here we develop an all-optical approach to probe synaptic plasticity at identified cerebellar synapses in awake, behaving mice. We developed and applied JEDI-2Psub, a genetically encoded voltage indicator with increased sensitivity around resting membrane potentials, to record subthreshold and suprathreshold activity in Purkinje cell (PC) dendrites while selectively activating their granule cell (GrC) inputs using optogenetics and their climbing fiber (CF) inputs using sensory stimulation. We measured synaptic potentials and complex spike signals across the dendrites of multiple neighboring PCs, enabling us to examine correlations in voltage signals within and between neurons. We show how pairing GrC activity with sensory-evoked CF inputs can trigger long-term plasticity of inhibitory responses in PCs. These results provide a blueprint for defining the rules for plasticity induction at identified synapses in awake animals during behavior.

Electrical signals are the internal language of neurons. Changes in transmembrane potential, expressed as synaptic potentials and action potentials, serve as the substrate for information received from and transmitted to other neurons[1]. Measuring these signals directly is essential if we are to understand information processing and storage in single neurons and networks of neurons[2]. In particular, the ability to measure synaptic dynamics in vivo over long timescales in genetically identified neurons is essential for testing longstanding theories of memory storage[3]. However, measuring the voltage dynamics of individual neurons in the intact brain is significantly challenging. Intracellular recordings using sharp microelectrodes and patch-clamp methods are typically brief, laborious and low-yield[4,5]. Therefore, systems neuroscientists commonly use two-photon microscopy in vivo[6,7] to record somatic calcium signals as a proxy for action potentials, and dendritic calcium signals as a proxy for synaptic activation[8]. The advent of a new generation of genetically encoded voltage indicators (GEVIs) suitable for two-photon microscopy[9–11] now offers the prospect of measuring voltage signals directly in networks of genetically identified neurons in vivo[12].

[1]Wolfson Institute for Biomedical Research and Department of Neuroscience, Physiology and Pharmacology, University College London, London, UK. [2]Department of Neuroscience, Baylor College of Medicine, Houston, TX, USA. [3]Systems, Synthetic, and Physical Biology Program, Rice University, Houston, TX, USA. [4]Centre for Developmental Neurobiology, King's College London, London, UK. [5]Department of Biochemistry and Molecular Pharmacology, Baylor College of Medicine, Houston, TX, USA. [6]Department of Electrical and Computer Engineering, Rice University, Houston, TX, USA. [7]School of Biomedical Sciences, The University of Hong Kong, Hong Kong, China. [8]Present address: Allen Institute, Seattle, WA, USA. ✉e-mail: jacques.carolan@ucl.ac.uk; m.hausser@ucl.ac.uk

Here we present an approach for measuring synaptic strength and plasticity induction in the awake behaving brain that harnesses the combination of an optimized GEVI, two-photon imaging, optogenetics and sensory stimulation. We achieved long-term and high-fidelity single-trial readout of postsynaptic voltage signals across the dendrites of cerebellar Purkinje cells (PCs) in the intact mouse brain, revealing voltage state-dependent modulation of complex spikes, and excitatory and inhibitory synaptic potentials triggered by optogenetic activation of presynaptic granule cells (GrCs). We show that pairing GrC activation with sensory-evoked climbing fiber (CF) signals—a canonical protocol for driving synaptic plasticity that is thought to be fundamental for cerebellar learning—triggers long-term potentiation of inhibitory synapses, and we assess its effects across the dendritic tree and across neighboring neurons. Our approach leverages a conventional two-photon microscope and therefore can be widely implemented in many neuroscience labs around the world. Our all-optical strategy enables rapid and non-invasive probing of plasticity induction and memory storage in networks of neurons in awake, behaving animals.

## Results

### Two-photon voltage imaging and optogenetics in vivo with JEDI-2Psub reveals subthreshold membrane potential dynamics in the cerebellum

To measure postsynaptic potentials in PCs, we first sought to enhance the sensitivity of JEDI-2P, a GEVI we recently reported[9]. We discovered that inserting a tryptophan between the GFP and the voltage-sensing domain (Fig. 1a) gave rise to larger single spike voltage responses ($\Delta F/F_0 = -34.1 \pm 6.8\%$ compared with $-23.4 \pm 3.5\%$, mean ± 95% CI, $n = 6$ and $n = 7$ HEK293A cells respectively) (Fig. 1b–e) with better photostability (Fig. 1f). Critically, this modification shifted JEDI-2P's voltage sensitivity toward more negative membrane potentials (Fig. 1g), producing 3.5x larger responses to voltage changes around the resting membrane potential (Fig. 1h), a critical feature for reporting postsynaptic potentials. We designated this indicator JEDI-2Psub, denoting its increased sensitivity for reporting subthreshold responses.

To probe synaptic plasticity at excitatory and inhibitory input synapses onto PCs, we developed an approach that (1) measures postsynaptic voltage signals in PCs via two-photon voltage imaging, (2) selectively activates GrC inputs via optogenetic activation, and (3) drives the CF pathway via sensory stimulation (Fig. 2a, b). Viruses expressing JEDI-2Psub and the red-shifted excitatory opsin ChRmine-mScarlet[13] were co-injected in Lobules V and VI of the cerebellar vermis. JEDI-2Psub was cloned downstream of the CaMKII promoter, producing selective expression in PCs. ChRmine was inserted in a Cre-dependent vector, enabling selective expression in GrCs in our Math1-Cre mice[14] (Fig. 2c, Supplementary Figs. 1, 2). Experiments were performed with awake, head-fixed mice on a running wheel and imaged using resonant scanning two-photon microscopy at 440 Hz (Fig. 2d).

We imaged Purkinje cell spiny dendrites expressing JEDI-2Psub at a depth of 50–100 μm below the cerebellar surface. They exhibited large spontaneous voltage transients ($\Delta F/F_0 = -31.2 \pm 4.4\%$, mean ± std, $n = 43$ cells across four mice) that could be readily observed in the raw signal (Fig. 2e, f; discriminability index d' = $5.9 \pm 1.3$; Supplementary Fig. 3). These events were characterized by a rapid depolarization, followed by a slower repolarization (FWHM = $10.5 \pm 1.8$ ms; Fig. 2g, h) and occurred at a rate of $1.3 \pm 0.4$ Hz (Supplementary Fig. 3), consistent with the properties of CF-driven complex spikes[15,16]. These events were variable in amplitude, with larger signals at more depolarized baseline membrane potentials (Fig. 2i), similar to electrophysiologically recorded complex spike signals in dendrites[17,18].

Next, we explored how PCs responded to sensory stimulation by applying brief airpuffs to the mouse whisker pad. Sensory stimulation evoked a diversity of voltage signals in PC dendrites, as expected from differing convergence of mossy fiber and CF inputs across the cerebellum[19] (Fig. 2j, Supplementary Fig. 4). Some PCs exhibited a largely inhibitory response (Fig. 2j, left), likely due to interneuron-mediated inhibition via the GrC pathway[20]. Other PCs exhibited a largely excitatory response (Fig. 2j, right), with a time to peak of $65.1 \pm 13.1$ ms (mean ± std, $n = 43$ cells across four mice; Supplementary Fig. 5), consistent with CF-driven sensory-evoked spike latencies in other cerebellar regions[17,21]. To confirm that sensory-evoked excitatory responses were CF-driven, we compared the spontaneous and sensory-evoked spike waveforms, finding a near-perfect match (Fig. 2k, l).

We next used optogenetics to selectively activate GrCs and drive parallel fiber inputs. Optogenetic activation of GrCs typically evoked graded hyperpolarizations in Purkinje cells that scaled with stimulus intensity (Fig. 2m, n). These hyperpolarizing events were initiated within a few milliseconds of optogenetic activation ($9.6 \pm 4.8$ ms, mean time from beginning of stimulus to 10% of peak amplitude) and had a time to peak ($85.4 \pm 24.4$ ms) consistent with in vivo results using electrical stimulation of granule cells[22]. They were significantly reduced by gabazine, confirming that they are mediated via $GABA_A$ receptors (Fig. 2o) and are thus feed-forward IPSPs triggered by optogenetically activated GrC excitation of molecular layer interneurons[20]. Their time course is likely shaped, in part, by the dynamics of GrC spiking[23], by the prolonged activation of interneurons by dense parallel fiber activation[24] (as has been observed in vivo under physiological conditions[25,26]) and by the kinetics of the GEVI. Optogenetic activation of GrCs sometimes triggered all-or-none depolarizations (corresponding to parallel fiber-evoked dendritic spikes[27]; Supplementary Fig. 5c), and occasionally triggered graded depolarizations (corresponding to parallel fiber EPSPs; Supplementary Fig. 5j–l). These responses had smaller amplitudes than spontaneous complex spike signals ($\Delta F/F_0 = -10.5 \pm 2.9\%$, mean ± std, $n = 7$ cells), peaked shortly after optogenetic activation (time to peak of $10.3 \pm 0.6$ ms) (Supplementary Fig. 5j, k), and were shaped by inhibition (Supplementary Fig. 5l). The time to peak of excitatory responses is comparable to results in the literature[28]; however we note that the combination of rapid termination of EPSPs by feed-forward IPSPs[20,29–32] and the 5–8 ms deadtime between LED onset and subsequent recording (see "Methods") resulted in optically evoked EPSPs being observed only rarely. Optical stimulation in preparations without opsin expression produced no responses (Supplementary Fig. 5i). Complex spike rate and amplitude did not significantly change between the beginning and end of our recordings (Supplementary Fig. 6a, b), and neither did they change between opsin-only mice and mice co-expressing the GEVI and opsin (Supplementary Fig. 6c, d), consistent with minimal activation of the opsin by the imaging pathway. Together, these results indicate our method can all-optically evoke and resolve IPSPs by combining optogenetic activation of GrCs and voltage imaging of PC dendrites, respectively.

### Spatiotemporal analysis of dendritic responses distinguishes synaptically driven and regenerative $Ca^{2+}$ events

Imaging dendrites of neighboring PCs would be a powerful method to probe the spatial relationship of voltage signals in the local network. To validate our method, we first examined whether we could detect synchronous complex spike signals in dendrites of adjacent PCs. Such events reflect anatomical and functional 'microzones' in the cerebellar cortex, driven by coupled presynaptic neurons in the inferior olive, that have previously been observed using electrophysiology[33,34] and calcium imaging[35,36]. Synchronized complex spike events were frequently observed in neighboring PCs (Fig. 3a, b). The mean peak latency and probability of sensory-evoked complex spike signals were highly correlated in neighboring cells ($R = 0.490$, $p = 1.15 \times 10^{-3}$ and $R = 0.590$, $p = 4.85 \times 10^{-5}$, respectively), $n = 49$ cells from 20 FOVs across four mice (Supplementary Fig. 7a, b), consistent with these PC pairs being part of the same microzone[36]. Furthermore, the temporal

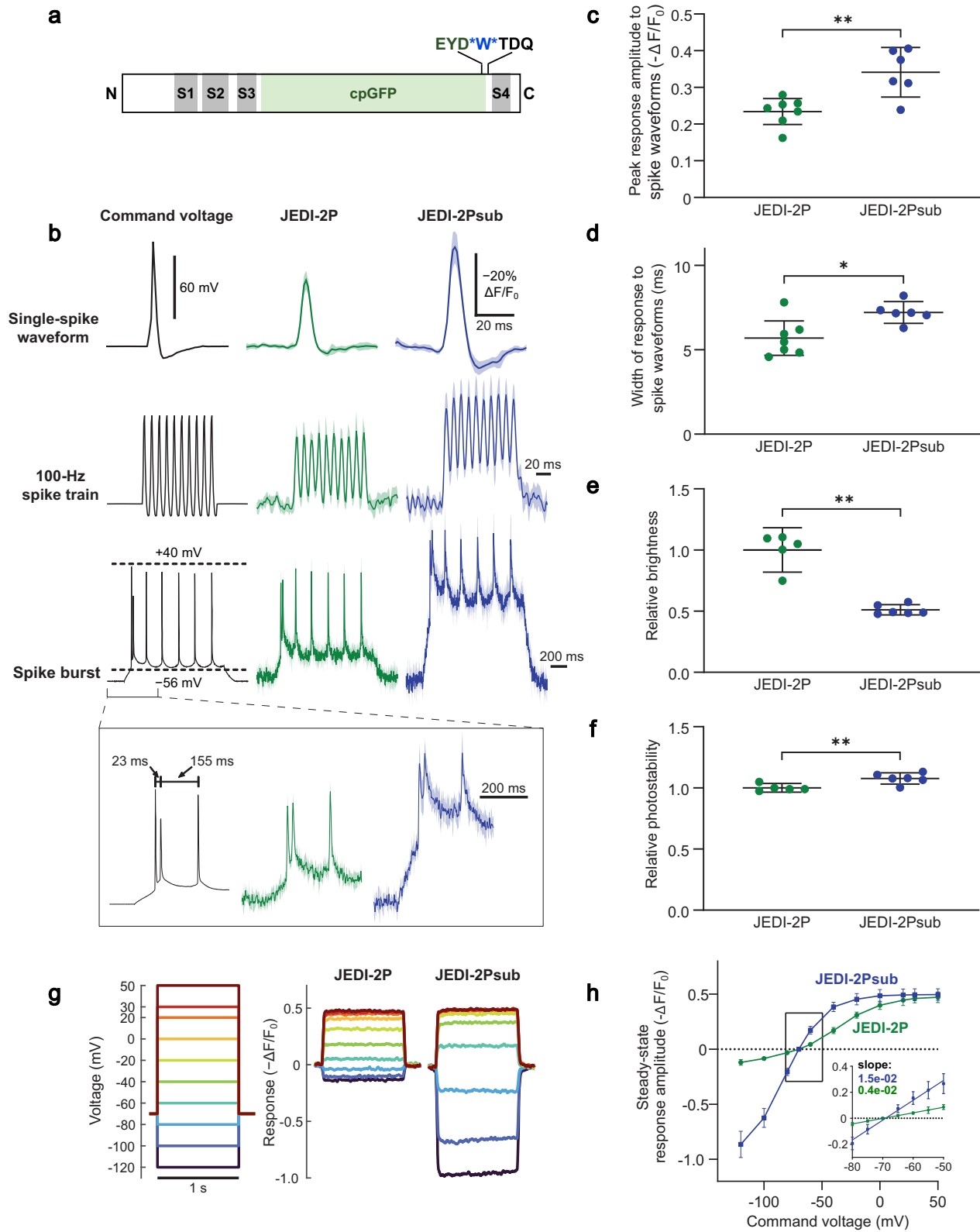

synchrony of spontaneous complex spike signals within the same microzone was very precise, with the mean width of cross-correlogram peaks being 6.4 ms (Supplementary Fig. 7h) (see "Methods"). However, the amplitude of complex spike signals in neighboring PCs, whether spontaneous or sensory-evoked, were not significantly correlated ($R = 0.286$, $p = 0.0695$ and $R = 0.219$, $p = 0.170$, respectively, $N = 20$ FOVs, $n = 49$ cells across four mice; Supplementary Fig. 7i, j),

suggesting that these signals are independently regulated in adjacent cells.

Next, we examined the relationship between IPSP signals in neighboring PC dendrites evoked by sensory stimulation and optogenetic stimulation of GrCs. Mean IPSP amplitudes in neighboring dendrites from different cells were tightly correlated (Fig. 3c, Supplementary Fig. 7c) ($R = 0.843$, $p = 4.55 \times 10^{-12}$ and $R = 0.807$, $p = 1.90 \times$

**Fig. 1 | JEDI-2Psub displays larger fluorescence responses to subthreshold voltage dynamics than JEDI-2P under laser-scanning two-photon excitation.** **a** An insertion of a tryptophan (blue) immediately after the circularly permuted GFP (cpGFP, green) in JEDI-2P created the modified sensor JEDI-2Psub. In this linear representation, gray bars represent the four transmembrane helices (S1-S4) of the voltage-sensing domain. **b** Mean fluorescence response to a single spike waveform of 2 ms full width at half maximum (top), a 100 Hz spike train (middle), and spike train on top of a subthreshold depolarization (bottom) ($n = 7$ JEDI-2P and $n = 6$ JEDI-2Psub HEK293A cells). **c** Quantification of the peak response of JEDI-2P ($n = 7$ cells) and JEDI-2Psub ($n = 6$ cells) to single spike waveforms (two-sided $t$-test $p = 3.40 \times 10^{-3}$). **d** Quantification of the width of the response of JEDI-2P ($n = 7$ cells) and JEDI-

2Psub ($n = 6$ cells) to single spike waveforms (two-sided $t$-test $p = 1.23 \times 10^{-2}$). **e** Brightness comparison. Values are normalized to JEDI-2P ($n = 5$ JEDI-2P and $n = 6$ JEDI-2Psub wells, two-sided Mann–Whitney U-test $p = 4.30 \times 10^{-2}$). **f** Photostability comparison. Values are normalized to JEDI-2P ($n = 5$ JEDI-2P and $n = 6$ JEDI-2Psub wells, two-sided $t$-test $p = 8.30 \times 10^{-2}$). **g** Mean fluorescence response to voltage steps from a resting membrane potential of −70 mV ($n = 7$ JEDI-2P and $n = 6$ JEDI-2Psub HEK293A cells). **h** Quantification of (**g**). Inset, response to subthreshold voltages. The inset displays the linear regression for each GEVI (JEDI-2P has a slope = 0.43 and $R^2 = 0.91$, JEDI-2Psub has a slope = 1.5 and $R^2 = 0.92$). Data are presented as mean values with 95% confidence interval. For all figures, *$p < 0.05$; **$p < 0.01$; ***$p < 0.001$.

---

$10^{-10}$ sensory evoked and optically evoked, respectively, $N = 23$ FOVs across four mice) with the correlation increasing with optical stimulation intensity (Supplementary Fig. 7k, l), as were the IPSP peak latencies ($R = 0.500$, $p = 8.71 \times 10^{-4}$ and $R = 0.474$, $p = 1.74 \times 10^{-3}$; Supplementary Fig. 7d, g), suggesting that neighboring Purkinje cells received shared inhibition from presynaptic interneurons[37,38].

We also examined voltage signals across the dendritic tree of individual Purkinje cells by segmenting the dendritic arbor into ~5 μm segments (Fig. 3d). We compared the homogeneity of the spatial distribution of the amplitude of sensory-evoked and spontaneous complex spike signals, and optogenetically evoked IPSPs (Fig. 3e) by calculating the coefficient of variation $|CV| = \sigma/|\mu|$ (see ref. 39) of the responses across each Purkinje cell dendrite (Fig. 3f). Complex spike signals, either sensory-evoked or spontaneous, were relatively uniformly distributed in the dendritic tree ($CV = 0.079 \pm 0.032$ (Fig. 3f, blue), $0.095 \pm 0.043$ (Fig. 3f, green), respectively, mean ± std, $n = 40$ cells across four mice) consistent with the widespread distribution of climbing fiber inputs in the dendritic tree and global depolarization[40,41]. In contrast (Fig. 3g, h), IPSPs showed a more heterogeneous spatial distribution ($CV = 0.130 \pm 0.061$; Fig. 3f, red), which presumably reflects the discrete and non-uniform activation of inhibitory synaptic inputs[42]. Complex spike signals and IPSPs were not correlated with baseline fluorescence, either at the level of individual neurons (Supplementary Fig. 6e, f) or dendritic segments (Supplementary Fig. 6g, h), indicating these variations are indeed physiological.

### In vivo plasticity induction induces LTP and normalizes activity across PC dendrites

Finally, we harnessed our approach to measure synaptic plasticity triggered by an all-optical induction protocol. We coactivated optogenetically recruited GrCs with sensory-evoked CF inputs—a classical pairing protocol for plasticity induction in the cerebellar cortex[43,44]—and optically monitored the synaptic efficacy of inhibitory inputs to PC dendrites via voltage imaging. We used a timing-dependent induction protocol where the activation of CF inputs is followed by the activation of GrCs inputs (separated by 150 ms, repeated 300 times at 1 Hz) (Fig. 4a). This induction protocol led to a robust long-term potentiation (LTP) of the IPSP signals (Fig. 4b, c; Supplementary Fig. 8a) which lasted for the duration of the experiment (40 min post-pairing) (Fig. 4d; pre−$\Delta F/F_0 = 7.9 \pm 4.1$ %, post−$\Delta F/F_0 = 9.9 \pm 4.6$ %; mean ± std; paired Wilcoxon signed-rank test $p = 8.80 \times 10^{-3}$, $n = 32$ cells across four mice). As controls, we omitted the pairing procedure and the potentiation was not observed (Fig. 4e; pairing group difference from the control group tested via a linear mixed-effect model with animal and FOV as random effects $p = 4.92 \times 10^{-6}$; control group $n = 32$ cells across two mice); we also reversed the order of the GrC and CF stimuli which also produced no significant potentiation (Supplementary Fig. 8b). Additionally, spontaneous complex spike signals remained stable over time (Supplementary Fig. 8c, d) providing a further control to demonstrate that the measured changes in IPSP amplitude were not an artifact of photobleaching due to continued laser illumination.

We next investigated which features best predict the magnitude of LTP following the pairing procedure and found that PCs with smaller baseline IPSPs exhibited larger LTP ($R = -0.543$, $p = 1.32 \times 10^{-3}$) compared with the control ($r = 0.349$, $p = 0.05$) (Fig. 4f). We also compared plasticity in neighboring PCs, and showed that the magnitude of LTP was highly correlated ($R = 0.611$, $p = 1.95 \times 10^{-3}$, $N = 9$ FOVs, $n = 24$ cells, across three mice) (Fig. 4g). Consequently, IPSP signals in neighboring PCs became significantly more correlated after plasticity induction (Mann–Whitney U-test $p = 4.40 \times 10^{-4}$; Supplementary Fig. 8e, f). Finally, we examined whether the spatial distribution of IPSPs in the dendritic tree of individual PCs changed after plasticity induction. LTP was accompanied by a reduction in the variability of the IPSP signal across the dendritic tree (Fig. 4h) compared with the control (Fig. 4i) (Mann–Whitney U-test, $p = 3.94 \times 10^{-3}$), even though the spatial distribution of the spontaneous complex spike signals in the same dendrites did not significantly change (Mann–Whitney U-test, $p = 0.204$ (Supplementary Fig. 8g, h)). These findings demonstrate our approach can be used to optically trigger and monitor plasticity across multiple dendrites and multiple PCs induced by a physiological pairing protocol in vivo.

## Discussion

We have introduced an approach that combines optogenetics with two-photon voltage imaging using a genetically encoded voltage sensor optimized for detecting subthreshold events, enabling optical control and readout of synaptic plasticity in individual dendrites in awake behaving mice. This method allowed us to compare subthreshold and suprathreshold signals within the dendritic tree and across neighboring neurons, enabling measurements of synaptic plasticity in single cells and neuronal networks. This strategy harnesses the key advantages of voltage imaging using genetically encoded voltage indicators: the ability to read out synaptic efficacy directly from the subthreshold signals and with cell type specificity, without having to rely on a proxy for synaptic strength such as changes in calcium concentration. JEDI-2Psub further enabled imaging of the same cells and subcellular structures repeatedly over long time periods, allowing synaptic efficacy measurements during plasticity induction and paving the way for their quantification during learning.

Our experiments revealed a type of inhibitory plasticity mediated by conjunctive activation of GrC and CF input, which adds to the growing recognition that inhibitory synapses are important substrates of plasticity[45]. This form of LTP could act to counterbalance LTP at excitatory synapses, and given that inhibition also regulates CF calcium signals[46], it may also act to limit future plasticity triggered by the CF. Future experiments are needed to explore the cellular mechanisms of this form of plasticity and the various parameters (e.g., relative timing of different signals[47–50]) governing its induction.

Our proof-of-principle experiments provide a blueprint for optical investigation of synaptic plasticity in the intact brain, harnessing the resolution and sensitivity of two-photon imaging with the speed and directness of voltage imaging. Our approach is also highly compatible with recent advances in high-speed microscopy[11,51,52]. Ultimately, by further refining this approach to monitor synaptic efficacy across the

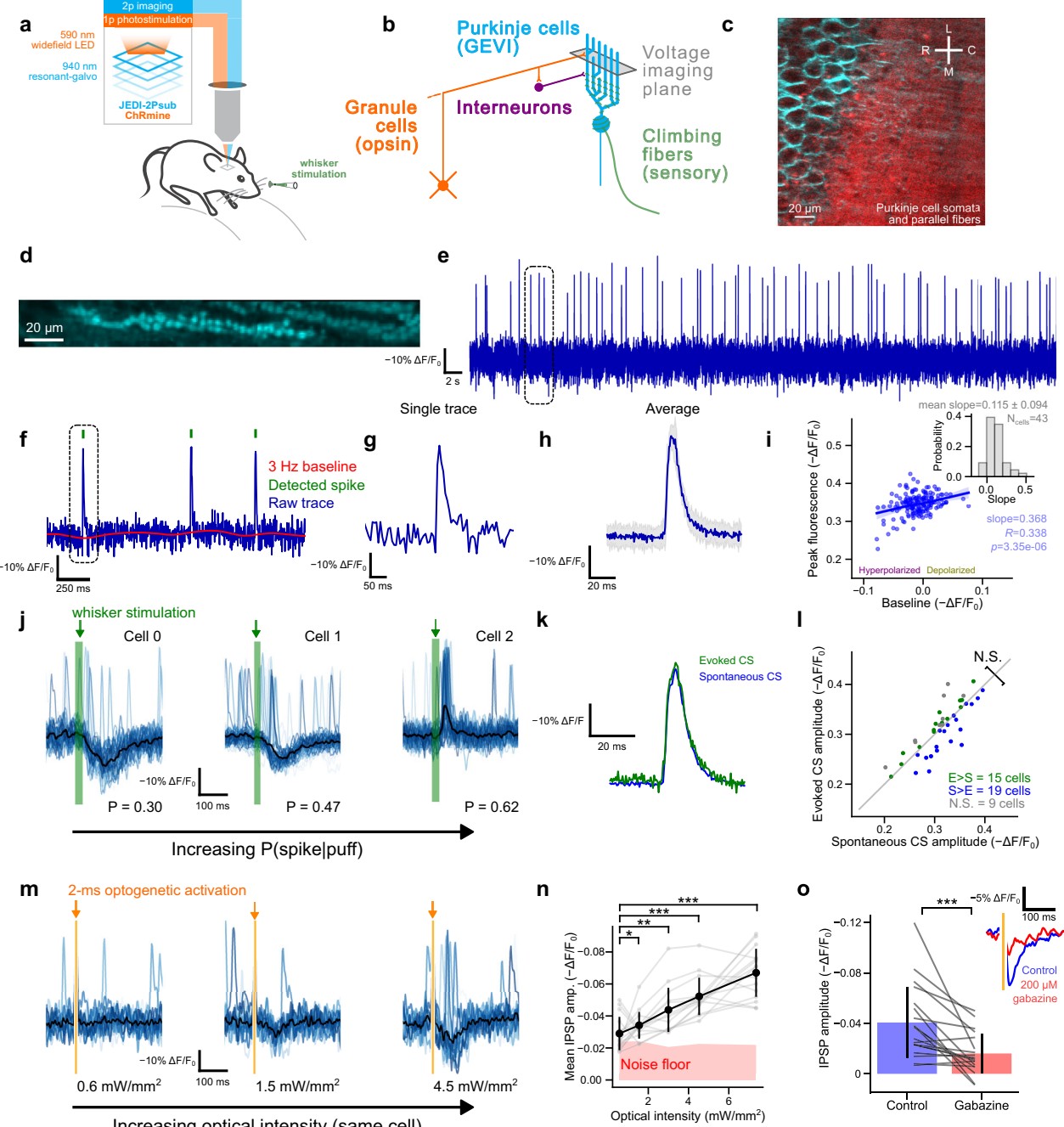

**Fig. 2 | In vivo two-photon voltage imaging and optogenetics reveal sub-threshold dynamics in Purkinje cell (PC) dendrites. a** Two-photon voltage imaging and optogenetic stimulation setup. **b** Cerebellar microcircuit. **c** Co-expression of the genetically encoded voltage indicator, JEDI-2Psub, in PCs (blue) alongside the opsin ChRMine expressed in granule cells (red). **d** Imaging of 254 × 30 pixel (208 × 24 μm) areas performed at 440 Hz. **e** 60 s unfiltered $\Delta F/F_0$ from a PC dendrite, imaged at 440 Hz. **f** 2 s unfiltered trace (blue) from the same cell, with baseline filter (red) and detected complex spikes (CS) signals (green). **g** Single complex spike from the same neuron. **h** The average of $n = 157$ spikes from the same neuron, up-sampled to 2 kHz (blue, gray ± 1 std). **i** Scatter plot comparing baseline fluorescence against peak fluorescence for each CS from a single PC (shaded area represents 95% confidence interval). Fitted by linear regression and tested via a two-sided Wald test. Inset: fitted slope across all cells. **j** A 25 ms air puff is applied to the whisker pad of the mouse at 0.5 Hz. Shown: the response of three different PCs

with the probability, P(spike|puff), of a puff response ($n = 60$ trials, mean response in black). **k** CS shape comparison between evoked (green, $n = 48$ spikes) and spontaneous (blue, $n = 243$ spikes). **l** Comparison of the mean spontaneous and sensory-evoked CS amplitude ($N = 4$ mice, $n = 43$ cells). Cells marked in green have a significantly larger evoked amplitude (35%), cells marked in blue have a significantly larger spontaneous amplitude (44%), and cells shown in gray are not significant (21%). Two-sided Wilcoxon signed-rank test $p = 0.844$. **m** Increasing the optical stimulation intensity increases the IPSP amplitude as shown across a single cell ($n = 19$ trials, mean response in black). **n** The mean IPSP response across multiple cells (black, error bars represent ±1 std, $n = 15$ cells and $N = 2$ mice). **o** Gabazine reduces the amplitude of optogenetically induced IPSPs (control $\Delta F/F_0 = 4.1 \pm 2.8\%$, post gabazine $\Delta F/F_0 = 1.6 \pm 1.6\%$, two-sided Wilcoxon signed-rank test $p = 1.26 \times 10^{-4}$, $N = 2$ mice, $n = 19$ cells, mean values ± 1 std). Inset: Gabazine control, single example (average of $n = 45$ trials).

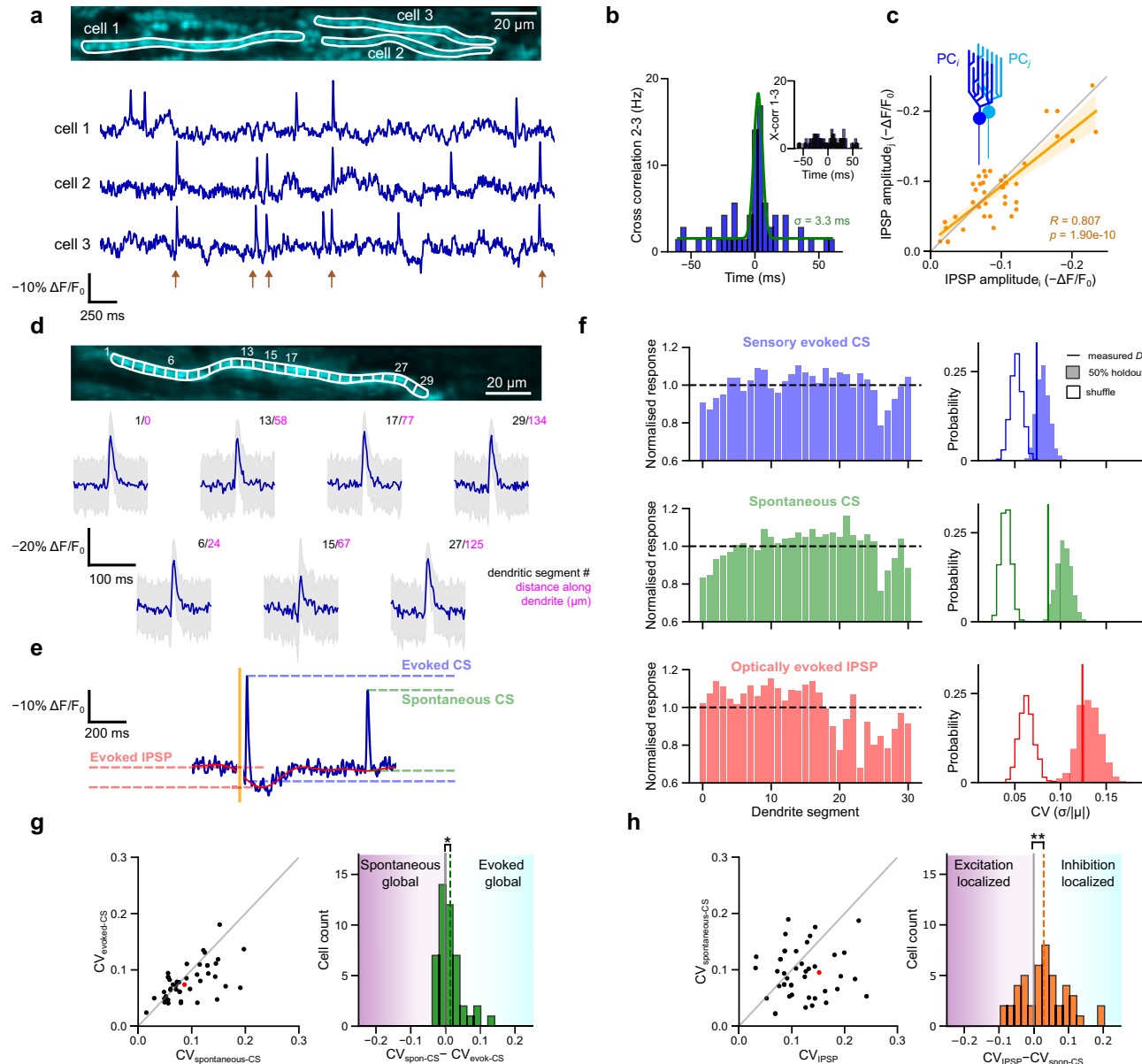

**Fig. 3 | Spatiotemporal analysis of PC dendritic voltage responses distinguishes synaptically-driven and regenerative Ca²⁺ events. a** Multiple PC dendrites are often recorded in the same FOV (top), revealing correlated CF activity (bottom) in the fluorescent traces (brown). **b** Cross-correlogram between PCs 2 and 3 shown in (**a**), overlaid with a Gaussian fit (green) indicating PC2 and PC3 are from the same microzone. Inset: cross-correlogram between PCs 1 and 3 showing no correlation, indicating they are from different microzones. **c** The magnitude of average optically evoked IPSP amplitude is plotted for all pairs of PCs (i,j) in the same FOV, showing highly correlated IPSP amplitudes between neighboring PCs ($R = 0.807$, $p = 1.90 \times 10^{-10}$ (fitted by linear regression and tested via a two-sided Wald test); $n = 23$ FOVs across four mice), which lies in the 99th percentile of correlation coefficients compared with a within-mouse shuffle (shaded area represents 95% confidence interval). **d** Each dendrite is segmented into ~5 μm sections (top), which

reveals variations in responses across the dendrite (bottom, average of $n = 380$ spikes blue, gray ± 1 std). **e** Features are extracted from the fluorescence traces on a trial-by-trial basis. **f** The normalized response of these features is then compared across the length of the dendrite (left), and the coefficient of variation (CV) is calculated for each feature of each cell (right). **g** The CV is compared for spontaneous complex spike (CS) and sensory-evoked CS (left). On average, spontaneous CS have a moderate, yet significantly larger CV than evoked responses, showing that the evoked responses have more uniform amplitudes (one-sided Wilcoxon signed-rank test, $p = 0.037$; $n = 40$ cells across four mice). **h** The CV is compared for optically evoked IPSPs and spontaneous CS. On average, IPSPs have a significantly larger CV, meaning IPSP responses are more localized than regenerative CS events (one-sided Wilcoxon signed-rank test, $p = 8.14 \times 10^{-3}$; $n = 40$ cells across four mice). Red point marks the PC shown in (**f**).

full dendritic tree[53,54], at single spines[55] and during behavioral tasks[50,56], the long-awaited goal[57–59] of linking plasticity at individual synapses with learning may finally be within reach.

## Methods

### Plasmid construction

Standard molecular biology techniques were used to assemble the plasmids. The sequence of the plasmids used was confirmed using

whole plasmid sequencing based on nanopore technology (Plasmidsaurus Inc.). JEDI-2P and JEDI-2Psub were cloned into a pcDNA3.1/Puro-CAG vector and used to characterize the kinetics of the GEVIs. A cyOFP1 reference protein was added to the C-terminal side of the GEVI via a GSSGSSGSS linker and used for all other voltage-clamp experiments.

Site-directed polymerase chain reaction (PCR) mutagenesis was used to construct JEDI-2Psub. The 20 μL PCR reaction mix contained

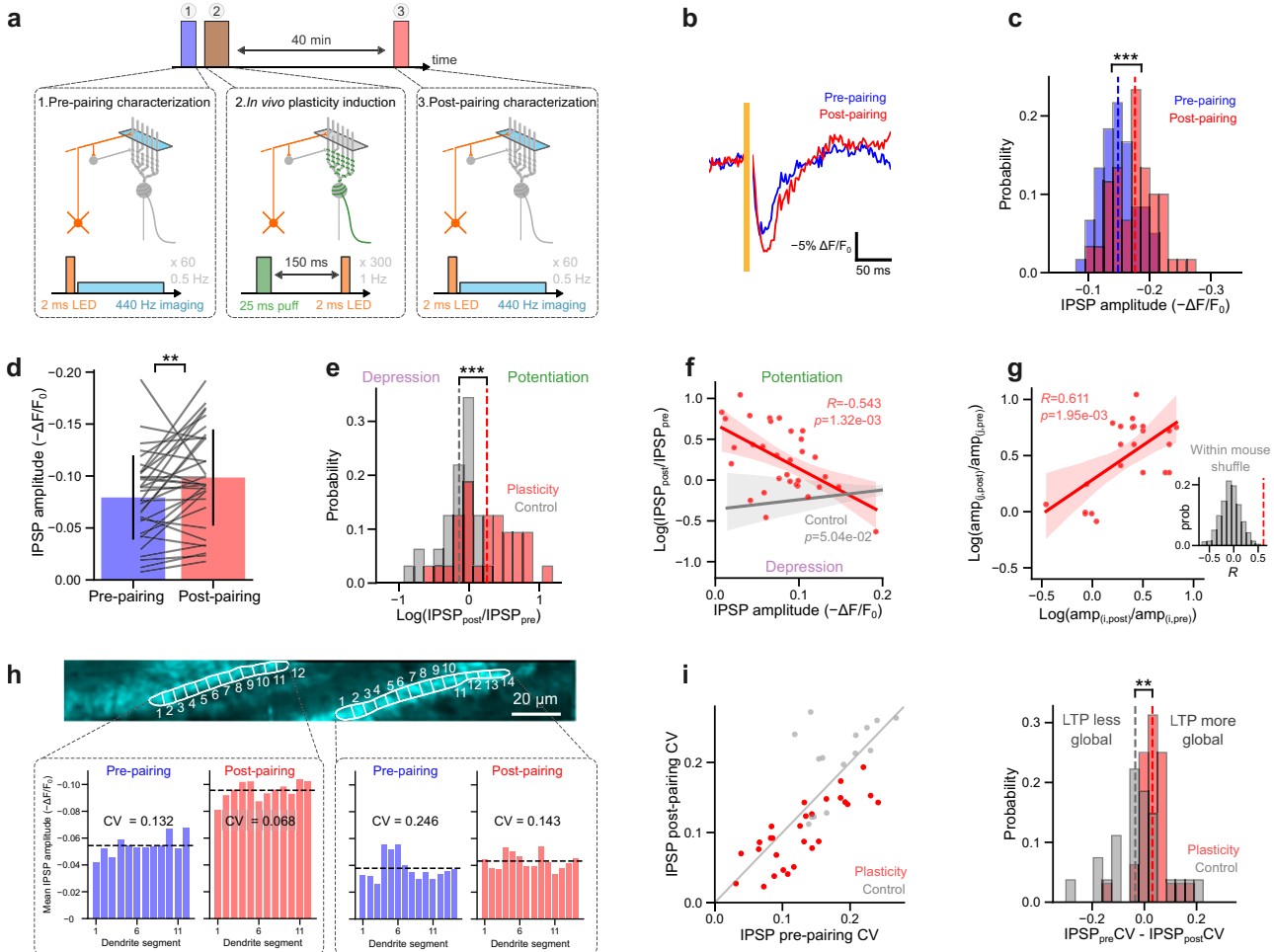

**Fig. 4 | In vivo plasticity induction triggers LTP of inhibitory responses and normalizes activity across PC dendrites. a** The experimental protocol comprises three stages: the first measures interneuron-PC responses by optogenetically activating GrCs and measuring evoked responses, the second pairs GrC activation with climbing fiber activation, and the third measures the post-pairing response. **b** Optical stimulation-triggered-average from a representative PC, pre- (blue) and post-pairing (red), $n = 60$ trials, displayed at 500 Hz. **c** A histogram of IPSP amplitudes for individual trials for the PC shown in (**b**), pre- and post-pairing ($\Delta F/F_0 = 14.8 \pm 2.8\%$ and $17.6 \pm 3.6\%$, respectively, mean $\pm$ std), two-sided Mann–Whitney U-test $p = 2.91 \times 10^{-5}$. **d** Mean IPSP amplitude change across all neurons ($N = 4$ animals, $n = 32$ neurons), pre- and post-pairing, two-sided Wilcoxon Signed-Rank test, $p = 8.80 \times 10^{-3}$. Error bars represent $\pm 1$ std. **e** A histogram of the log amplitude ratio for the pairing condition and the control condition (pairing protocol omitted). The two distributions were compared using a linear mixed-effect model with animal and FOV as random effects (two-sided Wald $t$-test $p = 4.92 \times 10^{-6}$). **f** A scatter plot of potentiation magnitude against pre-pairing IPSP amplitude, for the plasticity and control condition. The magnitude of the plasticity change is negatively correlated with the IPSP amplitude (red) but not for control (gray). Fitted by linear regression and tested via a two-sided Wald test. Shaded area represents 95% confidence interval. **g** For neighboring PCs (i,j) the magnitude of potentiation is plotted, showing a positive correlation between neighboring PCs (red), but not in a within-mouse shuffle distribution; inset: $n = 500$ shuffled correlation coefficients R in gray against the measured R (red dashed line; fitted by linear regression and tested via a two-sided Wald test; shaded area represents 95% confidence interval). **h** The change in IPSP amplitude across different dendritic segments is displayed for two PCs, for the pre- (blue) and post-pairing (red) conditions. Potentiation both increases the IPSP amplitude but decreases the variation in amplitude across the dendrite. **i** The coefficient of variation (CV) is plotted before and after pairing (left) for all neurons undergoing the plasticity protocol (red) and the control protocol (gray). The plasticity cells are significantly below the line of unity and below control (as shown in the histogram (right), two-sided Mann–Whitney U-test, $p = 3.94 \times 10^{-3}$).

1 µL forward primer mix at 10 µM, 1 µL reverse primer at 10 µM, 5 ng template plasmid, and 10 µL 2× PCR master premix (PrimeSTAR HS DNA polymerase, Takara). DNA was amplified using the following protocol: an initial denaturation step at 98 °C for 30 s; 30 amplification cycles of 98 °C for 10 s, 57 °C for 10 s, 72 °C for 1 min/kb of fragment length; a final extension step at 72 °C for 5 min. The pcDNA3.1/Puro-CAG backbone was linearized using the restriction enzymes NheI and HindIII. PCR products and linearized backbones were purified using gel electrophoresis and GeneJET Gel Extraction Kit (Thermo Fisher Scientific). PCR products were assembled in the vector backbone using the In-Fusion assembly system (In-Fusion HD Cloning Plus, Takara) according to the manufacturer's instructions. The In-Fusion reaction mix was transformed into commercial chemically competent bacteria (XL10-Gold, Agilent) with a transformation efficiency exceeding $5 \times 10^9$

CFU per µg DNA. Liquid cultures were inoculated with manually picked colonies, and purified plasmids were prepared using a minprep kit (VIOGENE Mini Plus plasmid DNA Extraction Kit Cat# GF2002) following the manufacturer's instructions.

## Two-photon screening
JEDI-2Psub was identified using the same high-throughput GEVI screening platform as previously described[9]. Brightness and photo-stability values were taken from the two-photon screening analysis. Each well had 4 non-overlapping FOVs that were 512 × 32 pixels. Images were acquired at 440 Hz using 920 nm set to 102 mW at the sample plane. FOV that had less than 300 pixels masked were removed from the analysis. To quantify the brightness of GEVIs independently of their expression level, green fluorescence of the GEVI was normalized to the

red fluorescence of the 3xGSS linked cyOFP1. Photostability was calculated as the normalized area under the curve of the green channel's fluorescence intensity (weighted by pixel count) for the duration of the screen (4000 frames). Normalization here refers to normalizing to the initial fluorescence of the trace.

## Preparation for voltage clamp

HEK293A cells (Thermo Fisher Scientific) were cultured at 37 °C with 5% $CO_2$ in growth medium, which contained high-glucose Dulbecco's Modified Eagle Medium supplemented with 2 mM glutamine, 100 unit/mL Penicillin, 100 mg/mL Streptomycin, and 10% fetal bovine serum (FBS). In preparation for electrophysiology experiments, 50,000 cells were transfected and plated on 30–70 kD poly-D-lysine-coated circular cover glass (12 mm #0, 633009, Carolina) two days before imaging. Following FuGENE® HD Transfection Reagent protocol (E2312, Promega), 100 ng of DNA and 0.3 μL of FuGENE mixed in Opti-MEM™ I Reduced Serum Medium (#31985, Fisher Scientific) were added to each circular cover glass in a 24-well plate (#3524, Corning® Costar® TC-Treated Multiple Well Plates). After transfection and plating, cells were cultured in the same growth media with 5% FBS. The day after transfection, the media was changed to fresh 5% FBS growth media.

Using a pipette puller (P1000, Sutter), glass micropipettes (1B150-F-4, World Precision Instruments) were prepared with a tip resistance of 2–6 MΩ. Micropipettes were filled with an internal solution of 115 mM K-gluconate, 10 mM HEPES, 10 mM EGTA, 10 mM glucose, 8 mM KCl, 5 mM $MgCl_2 \cdot 6H_2O$, 1 mM $CaCl_2 \cdot 2H_2O$, and adjusted to pH 7.4 with KOH. The micropipette was installed on a patch-clamp head-stage (CV-7B, Molecular Devices) and positioned by a micromanipulator (SMX series, Sensapex). A custom glass-bottom chamber based on Chamlide EC (Live Cell Instrument) was used to patch the cells. The glass-bottom chamber was made with a 24 × 24 mm #1 coverslip (D102424, Deltalab). Cells were continuously perfused with an external solution (110 mM NaCl, 26 mM sucrose, 23 mM glucose, 20 mM HEPES, 5 mM KCl, 2.5 mM $CaCl_2 \cdot 2H_2O$, 1.3 mM $MgSO_4$, titrated to pH 7.4 with NaOH) at a rate of ~4 mL/min with a peristaltic pump (505DU, Watson Marlow). An Axon Digidata 1550B1 Low Noise system with HumSilencer (Molecular Devices) was used to record voltage clamp data. Command voltage waveforms were compensated for the −11mV liquid junction potential of HEK293A cells. Recordings passed quality control and were included in the final analysis if the patched cell had an access resistance (Ra) equal to or smaller than 20 MΩ and a membrane resistance (Rm) larger than 10 times Ra both before and after the recording.

## In vitro imaging setup

A Ti-Eclipse inverted microscope with multi-photon capability (A1R-MP, Nikon Instruments) was used for voltage clamp experiments. Light was delivered to the sample plane through a ×40 objective (NA-0.95, CFI Plan Fluor oil immersion, Nikon Instruments). The scanning Stage SCAN^plus IM 130 × 85 for Nikon Eclipse Ti was used to control the position of the field of view. To support system automation, data acquisition and output boards (PXI-6229 and PXI-6723, National Instruments) were connected to the microscope computer through a PXI Chassis (PXI-1033, National Instruments).

## Two-photon imaging of JEDI-2Psub responses under voltage clamp

A Ti:Sapphire femtosecond laser (Chameleon Ultra II, Coherent) with a repetition rate of 80 MHz and a tuning range between 680 nm and 1080 nm was connected to the inverted microscope. A resonant galvanometer scanner was used to illuminate the cells with 940 nm light. Laser power was acousto-optically modulated to 20% (~61 mW) at the sample plane with the detector photomultiplier tube gain set to 20. Videos were taken at a resolution of 512 × 32 pixels with a frame rate of 440 Hz. Green fluorescence was filtered by a 525/50-nm (center

wavelength/bandwidth) filter and detected by gallium arsenide phosphide (GaAsP) photomultiplier tubes (PMTs).

Electrophysiological recordings were done at room temperature, and cells were held at −70 mV following a similar simulation protocol as previously reported for JEDI-2P (Liu et al.[9]). The protocol consisted of 5 × 2 ms FWHM AP waveforms at 2 Hz, 5 × 4 ms FWHM AP waveforms at 2 Hz, and 10 × 2-ms FWHM AP waveforms at 100 Hz. In addition, we also modified a burst of APs recorded from the adult mouse somatosensory cortex L5 pyramidal neurons to mimic APs on top of subthreshold depolarizations. The spike burst had a subthreshold depolarization of 24 mV (from −70 mV baseline voltage) and APs of 60–90 mV amplitude (from −56 mV subthreshold voltage). The APs in the spike burst were 3–4 ms FWHM. The last part of the protocol held the cells for 1 s at different voltage steps from a holding potential of −70 mV. Those voltage steps were −120, −100, −80, −60, −40, −20, 0, 20, 30, and 50 mV, followed by smaller steps around subthreshold membrane potential of −80, −85, −85, −60, −55 and −50 mV. After the AP waveforms, a 2-s interval at −70 mV was applied before the start of the voltage steps, followed by a 1.5-s interval between each voltage step. Voltage step traces were smoothed by a 47.6-ms moving average.

## Characterization of JEDI-2Psub's kinetics under one-photon illumination

The same inverted microscope as described above was used. Cells were illuminated with 475/28 nm light (Lumencor, Spectra III L light engine). A 5-band filter cube (#77015970, Semrock) was used for imaging. The 474/15-nm excitation band, the 493 nm long pass band of the dichroic, and the 515/30-nm emission band were used from the filter cube. A diaphragm was used to reduce the excitation spot to only one cell. Fluorescent changes were collected using a multialkali photomultiplier tube (PMT, PMM02, Thorlabs) installed on one of the side ports of the microscope. A MATLAB (R2023a 9.14.0.2206163 64-bit, The MathWorks) routine was used to set the PMT bias voltage to 0.99 and record the output voltage at 80 kHz using data acquisition and output boards. Electrophysiological recordings were done at 32–33 °C using a feedback-controlled inline heater system (inline heater SH-27B, controller TC-324C, cable with thermistor TA-29, Warner Instruments) to maintain the temperature in the perfusion chamber. Three 1 s depolarization pulses from −70 to 30 mV separated by 1.4 s at −70 mV were used to probe the kinetics of the GEVI (Supplementary Table 1).

A routine written in MATLAB was used to analyze the output voltage from the PMT. Data were downsampled to 20 kHz and division of the signal using a three-term exponential fit of the baseline was used to correct for photobleaching. The corrected signal was cropped 0.1 s before the estimated depolarization or the repolarization onset to 1 s after the estimated depolarization or repolarization onset. The exact onset timing was fitted together with other coefficients with either single-exponential ($F(t) = c + (k × \exp((t − t0) × \lambda)) × (t > t0) + k × (t ≤ t0)$) or dual-exponential ($F(t) = c + (k × \exp((t − t0) × \lambda) + k2 × \exp((t − t0) × \lambda2)) × (t > t0) + (k + k2) × (t ≤ t0)$) model where the t is the independent variable, F is the dependent variable, and the rest are the coefficients to be fitted. Among these coefficients, c describes the mean plateau fluorescence, k or k2 describes the relative ratio of each exponential component, λ or λ2 describes (minus) inverse of the time constant(s), and t0 is an offset indicating the exact event onset timing. JEDI-2Psub off-kinetics were best fit by a dual-exponential, where λ describes the off-kinetics and λ2 describes a slow photobleaching component not reported in Supplementary Table 1.

## Animals

All animal procedures were performed under license from the UK Home Office in accordance with the Animals (Scientific Procedures) Act 1986 and were approved by the local Animal Welfare and Ethical Review Board at University College London. We used a combination of male and female Math1-Cre mice[14] (line B6.Cg-Tg(Atoh1-cre)1 Bfri/J)

aged between three and five months. Mice were singly housed in individually ventilated cages (IVCs) equipped with environmental enrichment.

## Surgical procedures and virus injection strategy

Mice were implanted with a headplate, injected with virus and installed with a cranial imaging window in a single surgery session. Three hours before surgery, mice were injected with dexamethasone to reduce swelling during the procedure. Mice were then maintained under 1.5–2% isoflurane anesthesia for the course of the surgery and fixed into a surgical stereotaxic. Mice were injected with buprenorphine (1 mg/kg, subcutaneous, Vetergesic) peri-operatively for analgesia and then shaved, scalped and the trapezius muscles carefully retracted. An aluminum custom headplate with a 5 mm imaging well was secured over the cerebellar cortex using dental cement (Super-Bond C&B, Sun-Medical), centered on the posterior tip of the interparietal bone. A 3 mm craniotomy was then drilled above in the center of the imaging well to expose the cerebellar vermis.

A 1 µL virus mixture was then prepared containing the GEVI JEDI-2Psub (pAAV-CaMKIIa-JEDI-2Psub-Kv-WPRE, AAV2/1) diluted from stock to a titer of $5 \times 10^{11}$ VG/mL, and the cre-dependent opsin ChRMine (ssAAV-9/2-hEF1a-dlox-ChRmine_MRS_mScarlet_ERES(rev)_WPRE-hGHp(A)) diluted to a titer of $3 \times 10^{12}$ VG/mL. When injected at weak titers, the CaMKII promoter restricted JEDI-2Psub expression to PCs in a window of -10–20 days post-surgery, and the cre-dependent ChRMine gave rise to opsin expression in granule cells and parallel fibers. Four injections were then performed across lobules V and VI of the cerebellar vermis. At each location, -100 nL of virus solution was pressure injected at 400 µm and 200 µm below the brain surface, and we waited 5 min before the injection pipette was retracted. After injections, a 3 mm diameter single-paned coverslip was pressed into the craniotomy and secured in place first by a thin layer of cyanoacrylate (VetBond) and then by dental cement. For drug blocker experiments, custom-made coverslips with a $1.5 \times 0.5$ mm hole (filled Kwik-Cast) were installed. A rubber cone (RS Components, stock number 749-581) was attached to the headplate with dental cement (to minimize objective liquid evaporation), and the window was filled with Kwik-Cast to protect it during mouse recovery. Mice were given post-operative analgesia and allowed to recover for 7 days before viral expression was checked.

## In vivo imaging system

In vivo experiments were performed using a modified all-optical two-photon resonant scanning microscope (as described in ref. 60). JEDI-2Psub was excited via a Chameleon Ultra II laser (Coherent) at 940 nm and ChRMine was excited using a widefield 590 nm widefield LED (Thorlabs, M590L4) which was passed through interference filters (Chroma, ET 590/20) to minimize excitation leakage into the detection channels. Imaging laser and excitation LED were then combined onto a dichroic mirror and then propagated onto a ×16 water immersion objective (Nikon, N16XLWD-PF, 0.80 NA, 3.0 mm WD). Typical target imaging and excitation powers through the objective were 70 mW and 15 mW, respectively, corresponding to an optogenetic stimulation intensity of 8 mW/mm².

Fluorescent signals were then reflected back through the primary dichroic and then separated into red and green detection channels by a further dichroic mirror (Chroma, 575LP). To prevent damage to the imaging photomultiplier tube module, we used an electronically gated detector (Hamamatsu H11706-40) that was gated during LED photo-stimulation. LED stimulation, detector gating and sensory stimulation were all controlled via voltage signals from the microscope's onboard TTL voltage output. Electronic gating was set to 1 ms on either side of the LED control signal, which resulted in a 5–8 ms deadtime between LED onset and subsequent voltage recording (the specific length of this period was preparation- and FOV-dependent). Voltage signals, including LED stimulation, detector gating, sensory stimulation, two-photon imaging frame and y-galvo position, were recorded with a low-noise digital acquisition (DAQ) system (Axon Digidata 1550) with a 20 kHz acquisition rate.

## Image acquisition and processing

To capture voltage signals, we increased the frame rate by reducing the number of rows acquired by the resonant scanner to give a typical field of view size of $208 \times 24$ µm at 0.8 µm/pixel, giving an imaging frame rate of 440 Hz (Fig. 2d). This yielded an average of $1.64 \pm 0.84$ dendrites per field of view (mean ± std, $n = 50$ different fields of view from four mice). We were also able to record somatic signals; however, due to limitations in frame rate, we were only able to record somatic complex spikes (Supplementary Fig. 4). We typically restricted imaging sessions to 3 min to mitigate photobleaching across multiple sessions.

To extract signals from the same Purkinje cell dendrites, we used a Python-based data processing pipeline. The raw images were first registered using Suite2p[61] to reduce motion artifacts (Supplementary Fig. 1j). Motion artifacts in our experiments were low, with 95% of frames having motion below $X = 0.13 \pm 0.16$ µm and $Y = 0.11 \pm 0.11$ µm for sensory stimulation and $X = 0.09 \pm 0.02$ µm and $Y = 0.09 \pm 0.03$ µm for optogenetic stimulation (mean ± std across $n = 30$ recordings). Putative Purkinje cell dendrites were then manually identified based on morphological features from the mean registered image to generate a 2D cell mask $X$. A Gaussian filter was then applied $G(X)$ and the resulting mask was then normalized $N(G(X))$ such that the maximum intensity pixel equaled 1. This normalized cell mask was then elementwise cubed $N(G(X))^3$ to suppress low-intensity pixels and increase the signal-to-noise ratio. Each putative dendrite was then fitted to an intensity-weighted third-order polynomial and segmented into 4.8 µm segments. We then calculated the Pearson product-moment correlation coefficient between the fluorescence trace of every pair of segments across all putative dendrites in a given field of view (Supplementary Fig. 1k). Plotting this as a cross-correlation matrix revealed high-correlation clusters, presumably due to shared climbing fiber input to the same Purkinje cell. Dendritic segments were then clustered (Supplementary Fig. 1l) to yield signals from the same cells.

## ΔF/F calculation, baseline processing and spike detection

In the raw signal, spontaneous high-fidelity spikes could clearly be seen (Fig. 2e). To systematically detect these spikes, we first corrected for slow photobleaching effects by calculating $-\Delta F/F_0$ over a 2 s running window, where the minus sign corrects for the negative polarity of JEDI-2Psub (Supplementary Fig. 1m–o).

Our approach requires careful extraction of subthreshold baseline fluctuations and spikes. To do this, we first removed reductions in fluorescence from the raw signal due to the gating of the photo-multiplier tube during LED stimulation by interpolating between points on either side of the detector gating. We then low-pass filtered the resultant signal with a 5th-order Butterworth filter (typically 3–5 Hz) to yield an initial baseline signal and then smoothed the raw signal by applying a one-frame Gaussian filter. A spike was then assigned whenever a filtered point was greater than 3 std above the baseline, and the spike time was determined by the occurrence of the maximum point within a 25 ms window. Sometimes the width of the complex spike would visibly pollute the baseline signal. We therefore developed a two-step process where, for all datasets, we would remove spikes from the raw signal by interpolating between data points 10 ms before and 25 ms after the first pass detected spike, and then repeated the baseline calculation and spike detection on this clean signal. We found this to yield much more reliable baseline filtering and spike detection. For experiments that investigated voltage state modulation of complex spike amplitudes (Fig. 2i), we compare the peak fluorescence of every spike with the baseline fluorescence, defined as the $-\Delta F/$

$F_0$ low-pass signal that occurs at the same timepoint as the detected spike.

To calculate the detectability index $d'$, for every spike in a given cell, we extracted a signal data point and a background data point from the unfiltered trace. The signal was the spike amplitude ($-\Delta F/F_0$ at the spike time), and the background was a $-\Delta F/F_0$ value randomly selected from a window of $-250$ ms to $-200$ ms before the spike occurred. The mean signal $\mu_S$ and mean background $\mu_B$ were then used to calculate $d' = (\mu_S - \mu_B)/\sigma_B$.

To calculate the temporal synchrony between complex spike signals of neighboring Purkinje cells, we first calculated the cross-correlogram between adjacent spike trains using NeuroPyxels[62] with a bin width of 4 ms. We then used the unbiased estimation method of Stark and Abeles[63] to identify statistically significant data points in the cross-correlograms. For cross-correlograms with a peak greater than 10 counts, we calculated the temporal synchrony as the number of consecutive bins with statistically significant counts.

### Triggered average upsampling

For all stimulation and spike-triggered averages, we used upsampling techniques on the unfiltered $-\Delta F/F_0$ signal to increase the effective frame rate of the mean signal. For LED and sensory-triggered averages, we simultaneously recorded the voltage trigger signal and microscope frame onto the DAQ with 50 μs precision. The trigger signal and frame signal were not synced, which meant that the trigger occurred at a random point within the frame. This allowed us to calculate the true trigger-frame separation and thus upsample the resultant stimulation-triggered average (e.g., Figs. 2j, 4b). For spike-triggered averages, we fit the points in a 15 ms window around each detected spike to a Gaussian, which gave rise to a fitted spike offset and fitted spike width. This fitted offset was then used to upsample the spike-triggered average (e.g., Fig. 2h).

### Gabazine control experiments

To confirm our evoked negative voltage deflections were GABAergic in origin, we performed pharmacological blocker experiments. To allow access to the brain, we installed an imaging coverslip with a 1.5 × 0.5 mm hole for topical drug application. Once the animal recovered from surgery, we first imaged the control response to repeated optogenetic activation (2 ms at 0.5 Hz × 45 stimuli) across multiple fields of view (5–7), saving the x, y, z coordinates. We then replaced the objective liquid with a solution of gabazine (200 μM) and waited 10 min for the gabazine solution to disperse into the cerebellar cortex and repeated the same stimulation protocol across the same fields of view. We repeated this post-gabazine stimulation and imaging procedure every 10 min until 30 min post-gabazine application. Typically, the timepoint 20 min after drug application was used for analysis.

We carefully matched cells before and after gabazine application through a combination of comparing the cell morphology, segment-wise cross-correlation matrix, complex spike shape and firing rate. If a match could not be confidently made, that cell was excluded from the data set. We then extracted the IPSP signal on a trial-by-trial basis from each cell by taking the minimum filtered baseline signal in a 20–200 ms window post-optogenetic activation. In total, we measured $n = 19$ cells across two animals. The mean IPSP amplitude before ($\Delta F/F_0 = 4.1 \pm 2.8\%$, mean ± std) and after ($1.6 \pm 1.6\%$) gabazine application showed a statistically significant decrease in amplitude (Wilcoxon signed-rank test, $p = 1.26 \times 10^{-4}$).

### Plasticity protocol and analysis

For our plasticity experiments, we first characterized the pre-pairing response to optogenetic activation of granule cells and parallel fibers. We did this by giving repeated LED stimuli (2 ms at 0.5 Hz × 60 stimuli) while simultaneously recording from Purkinje cell dendrites. This was performed across multiple fields of view (typically 5–7) to increase the number of cells recorded in a given session.

Next, sensory stimulation was provided by applying an air puff (25 ms) to the mouse's whisker pad to activate the climbing fiber pathway, followed by a delay of 150 ms and then 2 ms optogenetic stimulation to activate the parallel fiber pathway. We repeated this 300 times at 1 Hz, to match classical pairing protocols in vitro[43]. We then waited 40 min to allow for plasticity effects to take place, and then repeated the optogenetic characterization protocol across all fields of view. As in our drug blocker experiments, we carefully match cells before and after pairing based on cell morphology, segment-wise cross-correlation matrix, complex spike shape and firing rate. We also registered the X, Y, Z coordinates of each field of view to prevent recording from the same cells multiple times. If cells could not confidently be matched, we excluded them from our analysis. In Supplementary Fig. 9a, b, we show five representative fields of view before and after our plasticity induction protocol alongside the mean squared error across all of our fields of view (Supplementary Fig. 9c). In total, four mice and 32 cells passed our selection criteria. We also performed control experiments, where mice went through the same pre- and post-pairing characterization, but the pairing protocol itself was omitted.

We extracted salient features on a trial-by-trial basis from each cell, including IPSP amplitude, taken as the minimum filtered baseline signal in a 20–200 ms window post-optogenetic activation, and spontaneous complex spike amplitude. Critically, the exact same data processing parameters were used for each cell pre- and post-pairing. In control data, there was a slight depression in IPSP amplitude, presumably due to calcium influx from the optogenetic activation protocol (Wilcoxon signed-rank test $n = 32$ cells, $p = 1.06 \times 10^{-3}$). In the pairing data, there was a significant potentiation effect (Wilcoxon signed-rank test $n = 32$ cells, $p = 8.80 \times 10^{-3}$) which was not observed in spontaneous complex spikes, serving as a further control.

### Statistics and reproducibility

No statistical method was used to predetermine sample size. Sample sizes of numbers of neurons and mice were similar to other in vivo two-photon voltage imaging studies[9,10]. Mice with poor viral co-expression of the GEVI and opsin were omitted from the study. PC dendrites that did not respond to optogenetic activation were not recorded. Experiments were not randomized, and the Investigators were not blinded to allocation during experiments, but were blinded to the plasticity timing condition during the data analysis phase.

## Data availability

The datasets supporting the findings of this study are available from the corresponding authors upon request. Source data are provided with this paper. JEDI-2Psub expression plasmids have been deposited to Addgene (plasmid #247186 and #247223). Other reagents will be shared upon request to F.S.-P.

## Code availability

Custom code used for post-processing Purkinje cell dendrites and data analysis has been deposited online: https://doi.org/10.5281/zenodo.16955138, https://doi.org/10.5281/zenodo.16955331.

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

## Acknowledgements

We thank Mehmet Fisek, Dustin Herrmann, Alex Prodan and Arnd Roth for comments on the manuscript, and Zihui Zhang for helpful discussions. We are grateful to Soyon Chun and Caroline Reuter for technical assistance. This work was supported by the Wellcome Trust (M.H., PRF 201225/Z/16/Z, D.K., CDA 225951/Z/22/Z), the European Research Council (M.H., AdG 695709), the Biotechnology and Biological Sciences Research Council (J.C., BB/W010623/) and the Chan Zuckerberg Initiative. We thank Dr. Benjamin Arenkiel, Joshua Ortiz-Guzman, and Zihong Chen at the Intellectual and Developmental Disabilities Research Center (IDDRC) Neuro Connectivity Core for packaging the GEVI AAV; this Core is supported by NIH grant P50HD103555 and the Charif Souki Fund. F.S.-P. is supported by National Institute for Health grants (U01NS133971, R01EB032854, R01NS146078, R01NS146023, RF1N-S128901, R61CA278458, R01NS136027), a Welch Foundation grant (Q-2016-20250403), the McNair Medical Foundation and a Vivian L. Smith Endowed Professorship in Neuroscience.

## Author contributions

J.C. and M.H. conceived the project. M.A.L., X.L., F.S.-P. designed, developed and tested the JEDI-2Psub construct. J.C. performed surgeries, performed in vivo imaging experiments and analyzed data; M.A.L. and X.L. performed in vitro experiments and analyzed data. M.B. and D.K. performed histology. B.C. provided input on experimental design. J.C. and M.H. wrote the paper with input from all authors.

## Competing interests

F.S.-P. holds a US patent for a voltage sensor design (patent #US9606100 B2) that encompasses the GEVI reported here. The other authors declare no competing interests.
