## [Transparent Peer Review file · Nature Communications]

All-optical voltage interrogation for probing synaptic plasticity in vivo

Corresponding Author: Professor Michael Häusser

Version 0:

Reviewer comments:

Reviewer #1

(Remarks to the Author)

The authors present a nice proof of principle for sub-cellular voltage measurements in response to sensory and neural stimulation, and compelling evidence for measuring stimulus-induced synaptic changes in vivo in the cerebellar cortex. The techniques are cutting-edge, the analysis and presentation are mostly clear, and the claims are generally appropriately matched to the data. Here are some suggestions.

The elephant in the room is that the authors apparently only rarely detect EPSPs in Purkinje cells following the presumably enormous (1p) excitation of thousands of granule cells, possibly even the majority of all GrC inputs to these PCs. I can find one EPSP example (S4J) and no quantifications of the respective frequency of detection of IPSPs vs EPSPs.

- I would appreciate some quantification of: in what fraction of cells/dendritic compartments/GrC stimulus presentations/mice the authors actually detect EPSPs and IPSPs.

- The authors should then also quantify any changes in the frequency of observed EPSPs in the gabazine condition

- Beyond the data analysis, at the moment, the text “just lets it slide,” which for my taste is not a great strategy. Some acknowledgement of this strange result, and some plausible explanations for what might be happening, are the minimum that would be appropriate here. Of course I agree with the authors’ interpretation of the origin of the GrC-evoked IPSPs as reflecting feedforward inhibition through MLIs, but this does not “explain away” the dearth of preceding GrC->PC EPSPs, which must come 1 synapse before the GrC->MLI->PC IPSP.

- Can the authors quantitatively exclude that this results from either (a) the scan rate (440 Hz) or (b) the GEVI (JEDI) being too slow to detect most EPSPs that must precede IPSPs? I.e., they are mostly getting missed/summed/blurred into a single IPSP?

- Can the authors quantitatively exclude that fast early EPSPs are missed due to the PMT gating at 1p laser pulsing (e.g., stability/kinetics of the PMT signal recovery post-gating)? The authors cite (p9LastPar) some kind of issue with PMT signals after gating, so this seems to be a problem?

- If the authors can exclude the above technical concerns, do they favor some biological explanation, e.g., globally jacking up GrCs causes Golgi cells to clamp down so there’s not much net effect (but then how to reconcile with the MLI result?).

My other major technical concern that is basically unaddressed is brain motion. The authors cite (P9par3) a FOV 24um “tall.”

- The authors should provide numbers on the peak magnitudes of X-Y motion artifacts (i.e., taken from the image registration algorithm output) during sensory and optogenetic stimulation. Even better would be to provide these numbers from larger FOV recordings so we can be certain that the motion is fully compensated rather than being “capped” at 24um.

- They should thereby quantitatively demonstrate that the membranes they’re recording from aren’t simply disappearing out of the 24um FOV

- They should also compare the measured motion magnitude to the expected zPSF optical section thickness to demonstrate that these very small structures aren’t vanishing from the optical section, or worse, being replaced with other small membrane segments, ideally using the density of labeled membrane segments as the relevant parameter.

It sounds (P11Par2) like it was not that easy to register dendrite identity 40 minutes after the “plasticity induction” protocol. Thus, it would be appropriate to provide at least examples of the first and second images of the same segments, and better yet a distribution across experiments of some image similarity metric, to demonstrate that this procedure was successful.

A few other points

The authors cite (P9par3) “50 different fields of view from 4 mice.” I’d like to hear how the authors confirmed that they didn’t come back to the same place twice when collecting ~12 fields per animal.

Fig. 1, N would ideally be more than 3.

P4Par3: “...spatial distribution...which presumably reflect” should say “reflects” or “reflected”

P7par3 "photomultiplier tubes set to 20" 20 what?

Reviewer #2

(Remarks to the Author)

Recent advances in voltage imaging technology allow optical access to the intracellular voltage dynamics in various animal models. However, voltage imaging at subcellular resolution and particularly the optical recording of synaptic potentials are still challenging. In this work, Carolan et al demonstrate voltage imaging from superficial Purkinje cell dendrites in the cerebellum of awake mice. By inducing presynaptic activity using either optogenetics or sensory stimulation they also recorded postsynaptic potentials at dendritic resolution and managed to detect inhibitory plasticity following LTP protocol. The key technological progress enabling these nice measurements was the development of a new variant of the GEVI Jedi-2P which shows higher sensitivity at the hyperpolarized regime and thus allows high-fidelity detection of synaptic potentials. This is a timely work, the data is sound, and the paper is well-written. Below I have a few comments that I hope could further improve the manuscript.

1. Amplitudes of fluorescence events (i.e. dF/F) should be treated with care, particularly in cases of comparisons between different ROIs. Specifically, dF/F is highly sensitive to the baseline fluorescence and its amplitude is therefore quite meaningless. This particularly applies to the cross-correlation analysis between the complex spikes (FigS5I-J), and to the comparison of IPSP amplitudes between cells (Fig3C). My suggestion would be to normalize the activity of each cell to the peak of the spikes whose underlying voltage can be assumed to be equal between different cells, and then repeat all those comparisons and update the conclusions accordingly.
2. The previous point is strengthened by the data in Fig.3F which shows variability in the spike amplitude along the dendrite. I'm pretty sure that this is not physiological, and it is likely the result of variability in the indicator expression along the tree. It thus questions the conclusion that IPSP amplitude is variable. Also here, I suggest normalizing the IPSPs to the spike amplitude in each segment and then reconsidering the claims in Fig.3F-H (and also removing the two top panels in Fig.3F).
3. The exact same comment applies also for Fig.4H

Minor comment:

One advantage of this work is the usage of a conventional 2P microscope which is available in many neuroscience labs. I think that this point should be mentioned in the introduction. On the other hand, many new high-speed 2P methods have been published in recent years, and techniques such as AOD-based random access microscopy, FACED, and SMURF could significantly improve and upscale these measurements in space and time. I suggest mentioning this point in the discussion and citing the relevant works.

Reviewer #3

(Remarks to the Author)

Remarks to the Author:

This work by Carolan and colleagues describes an optical approach by pairing one-photon bulk stimulation of granule cells with two-photon genetically targeted voltage imaging of purkinje cells to study the synaptic transmission and plasticity between granule cells and purkinje cells. For that, they develop one genetically encoded voltage indicator, JEDI-2Psub, which is expressed in purkinje cells, and use an optogenetic actuator, ChRmine, which is expressed in granule cells using a genetic mouse line (Math1-Cre). The authors are able to resolve complex spike signals and granule-cell optogenetic stimulation-induced potentials in purkinje cell dendrites.

Simultaneous optical imaging and stimulation is an exciting field, and the authors previously had multiple works on simultaneous optical Ca^{2+} imaging and optogenetic stimulation with careful characterization and discussion of the optical crosstalk. Here however, in this study, with different tools in demand of much higher power, the authors did not characterize or discuss the potential optical crosstalk (activation of the optogenetic actuator, ChRmine, when imaging JEDI-2Psub with $>60mW$ power and with high-speed scanning).

On the biological side, the authors can resolve granule-cell optogenetic stimulation induced potentials in purkinje cell dendrites, which is impressive. However, given the long latency between the optogenetic stimulation and optically induced response, the authors should discuss why and whether this is a truly synaptic effect or multi-synaptic effect (because this will affect the interpretation of the plasticity results). Also given the potential optical crosstalk, it is unclear whether granule cells are already depolarized, or the postsynaptic dendrites maybe already depolarized or hyperpolarized.

Finally, the authors pair granule cell optogenetic stimulation with sensory stimulation for studying plasticity. Using canonical plasticity induction protocol, the authors are able to detect changes in granule-cell optical-stimulation induced voltage signals in purkinje cells. However, the authors did not address potential intrinsic excitability change during the process. As a result, the interpretation of the plasticity experiments is not convincing. Furthermore, these plasticity experiments also need further controls. And because of the multiple synapses involved, the interpretation needs to be precise in terms of whether it is plasticity of inhibitory synapses.

Overall, this work is exciting. Because of the opportunity for high interest, it is essential that the authors be held to a high standard in terms of characterization of the optical tools and interpretation of the plasticity results. Provided changes and/or new data to address the concerns, I would support the publication of the work.

My major points are detailed below.

1. Genetically targeted voltage imaging with JEDI-2Psub:

The ability to detect voltage in genetically identified cells in vivo is impressive. The author should acknowledge prior work of genetically targeted voltage imaging in purkinje cell dendrites (Gong, Y., ..., & Schnitzer, M. J. 2014. Nature communications).

Secondly, the authors noted that they could detect voltage signals well in the soma. Could the author differentiate simple vs

complex spikes? This needs to be discussed.

Thirdly, how does the non-linearity affect the detection of subthreshold excitatory inputs to the purkinje cells? It is somewhat surprising that in fig S4H, most optogenetic stimulation induced excitatory responses are similar amplitude as complex spikes. Are they potentially big EPSPs? How many cells show clear EPSPs with a smaller amplitude compared to complex spikes as in fig S4J?

2. Optical crosstalk between imaging and stimulation:

Does the imaging light used to image JEDI-2Psub activate the optogenetic actuator ChRmine? ChRmine is a very sensitive opsin. Neurons expressing ChRmine showed notable depolarization with high imaging power (for instance, see reference: Sridharan, Savitha, et al., Neuron. 2022, Marshel, James H., et al. Science 2019.). This crosstalk is larger in a smaller field view with high scanning speed. This crosstalk effect is important for the interpretation of the results. For instance, would the imaging light already deplete the synaptic vesicles in granule cells? Would this already affect the synaptic inputs and resulting resting potential of the purkinje cells? Would the authors see a change in the baseline of voltage recording during the first few seconds of turning on the imaging light? Or change of spontaneous spiking rate during continuous imaging?

3. Optogenetic recruitment of granule cells:

It is hard to see the granule cells expressing ChRmine in Fig 2C or Fig S1. Could the authors show clearer images? How many granule cells are on average recruited during optogenetic stimulation? What is the latency between optogenetic stimulation onset to the optically induced EPSP, IPSP, and spikes? The reported latency in Fig. S4J is very long, presumably involving multiple synapses. Could the authors explain how they quantify the latency and why it is so long? Potentially this involves circuit effect, eg, multiple synaptic interactions among granule cells or the golgi cells or other cells. This should be discussed because it will affect the interpretation of the plasticity results in terms of which synapses are modulated.

The experiments with gabazine application are convincing that the observed hyperpolarizing events are via gabaergic synapses. With gabazine, would the authors see more optogenetically induced EPSPs and a shorter latency? If not, the authors should discuss why.

4. Correlation analysis:

The spatiotemporal analysis is nice. Regarding the correlation of IPSP, if a lot of granule cells are recruited during the optogenetic stimulation, the neighboring purkinje cells will have high correlations of IPSPs even when they do not receive shared inhibition from presynaptic interneurons. The authors may consider analyzing the correlation under different stimulation light intensities, like in Fig 2M.

5. Plasticity interpretation:

The authors include controls omitting the pairing procedure. What about omitting either activation of granule cells or sensory inputs? Would the observed change be due to the animal learning of the repeated sensory stimulation? Or could it be from the repeated optogenetic stimulation of granule cells?

Secondly, could the intrinsic excitability change during the process? Does the baseline of the voltage in purkinje cells change?

Since the observed IPSPs are at least through two synapses, one from granule cells to the inhibitory neurons, and one from the inhibitory neurons to the purkinje cells, can the author discuss where the potentiation occurs? Could it be the potentiation from granule cells to the inhibitory neurons? If not, the authors should discuss why. Otherwise, the authors should delete the claim that this is the plasticity of inhibitory synapses.

Fourthly, is there any activity change in response to sensory stimulation after the plasticity induction?

Lastly, the authors mentioned long time voltage imaging multiple times. They should specify how long do they perform the voltage imaging for each session. Is it only three minutes (as indicated in the methods part)? If so, it is not 'over long time periods'. Also, is the voltage imaging conducted during plasticity induction? What are the voltage dynamics during the induction process? This is important because potentially one can correlate the activity with resulted plasticity.

6. The manuscript emphasizes 'behavior' multiple times. Could the authors quantify the behavior during optogenetic stimulation and the plasticity induction process?

Minor:

1. Fig 2G: 'raw single spike' -> single complex spike

2. Fig 2J: how many trials average?

3. Fig 4B, could the authors show the single trial IPSP data? What is the noise level like compared to the observed difference?

4. If single trial IPSPs could not show the potentiation effect, the authors should delete the claim of 'single-trial read-out of synaptic plasticity' in discussion line 2.

5. The authors should acknowledge prior all-optical work to probe synaptic plasticity during behavior (Deisseroth 2023 Cell, 2024 Science).

Version 1:

Reviewer comments:

Reviewer #1

(Remarks to the Author)

Overall, the authors have addressed most of the suggestions well. I still have a significant lingering concern about the EPSP/IPSP business that I hope the authors can address simply, as it's unlikely I will be the only one to notice this. The authors state that most optogenetically-evoked EPSPs in PCs turn on with too short a latency to see due the ~10ms PMT gating signal recovery time—fine. They *also* state that the EPSPs are too *brief* to see even the decay from peak, and then assert that this is in turn due to the EPSP being quickly drowned by the overwhelming recruitment of interneurons—also fine.

Fig. S4L shows a control/gabazine example that makes this case well, and so far the argument holds together. The problem is, in evidently almost all cases, their IPSPs are way way slower than this – with a mean optically-evoked latency of almost 100 ms (!), Fig. S6J. This presents multiple problems that need to be acknowledged/addressed somehow:

- (1) These IPSPs cannot be the simple feedforward inhibition that curtails GrC responses—they are off by an order of magnitude. The difference in IPSP timing between Fig. S4L-gabazine (ipsp peak <20 ms) and the more typical Fig. S6J (peaks 80-100ms) makes this very clear.
- (2) Most of these IPSPs must be a bunch of synapses away from the GrC activation, perhaps even some secondary motor response to the GrC activation?
- (3) Again, I do not think it helps to just 'let it slide' in the text. The authors should probably disaggregate their IPSPs into a story that makes sense: (1) IPSPs that could plausibly be monosynaptic feedforward inhibition directly caused by GrC optogenetic activation (and how often were these actually seen??), and (2) then...something else, which should include some plausible physiological proposal for what the something else is, so readers don't start asking themselves the same questions I posed above.
- (4) this also assumes (hopefully) that the authors actually have evidence for the monosynaptic feedforward inhibition, since such fast IPSPs are also the authors' proposed explanation for the absence of EPSPs in the data (that they get shut down within <20 ms).
- (5) Is there another explanation for all of this? I started looking in more detail at how the authors did the optogenetics, and I see that they used ChRmine virus in adult Math1-Cre mice. No histology is shown. I apologize that I did not comment on this in the first round, but it is important to nail down. To my knowledge, *Math1-Cre has no cre expression in adult cerebellar granule cells* (10.1016/j.neuron.2005.08.028)—it expresses in progenitors and thus is only suited for granule cell expression when used in *crosses*. This suggests that whatever neurons the authors have successfully infected here is somehow due to cre-independent leak expression. This then raises the question of what else is expressing ChRmine. I am not normally a stickler for providing histology, but since the authors' strategy should in principle not have worked at all, it seems to be very necessary. The authors should ideally provide histology that includes the cerebellum and some major pre-cerebellar nuclei like the pons. My point in bringing this up is in trying to understand the confusing above results (lack of EPSPs and also possibly lack of short-latency monosynaptic feedforward inhibition IPSPs). One possibility is that the authors are not only (or not primarily) activating granule cells.

Reviewer #2

(Remarks to the Author)

The authors have addressed my comments well and I don't have further questions.

Reviewer #3

(Remarks to the Author)

Remarks to the Author:

The authors have made an effort to enhance the manuscript. The primary biological claim is that pairing optogenetic stimulation of granule cells with sensory stimulation can trigger long-term potentiation of inhibitory synapses. However, without further analysis and appropriate controls, this claim should be tempered throughout the paper.

Major:

1. Latency of IPSP Response:

The long latency between optogenetic stimulation and IPSP (85.4 ± 24.4 ms) should be explicitly stated in the main text, along with a discussion of its implications. Feed-forward inhibition (mentioned on lines 113 and 122) typically occurs within 1–2 ms, as cited in Mittmann et al. (2005):

"We have demonstrated directly that FFI is activated by PF input in just over 1 ms on average, with delays of less than 1 ms observed in some neurons."

Given this, how can feed-forward inhibition explain the observed ~85 ms latency? This discrepancy needs further clarification.

2. Noise Level in dF/F Measurements:

The noise level should be clearly specified in dF/F format in both the main text and Figures 2 and 4. From a visual assessment of Fig. 2E and F, and the d' value (line 93), the noise level appears to be around 3–5% dF/F. If this estimation is accurate, how is it possible to detect the single-trial plasticity depicted in Fig. 4, where the average increase is only ~2% dF/F—lower than the mean noise level? The authors should address this concern explicitly.

3. Imaging Rate and Temporal Resolution:

In line 148, please specify the imaging rate. If the imaging rate is 440 Hz, how can it reliably detect temporal synchrony of 1.93 ms—less than the duration of a single frame? This should be explained in more detail.

4. Context of Voltage Imaging Advances:

While the study effectively demonstrates voltage imaging at subcellular resolution, recent advances using one-photon voltage imaging have also achieved subcellular resolution and enabled all-optical interrogation of synaptic plasticity when combined with optogenetics. These works should be at least acknowledged: Park et al., Nature Communications (2025), Wong-Campos et al., bioRxiv (2023), Deisseroth, Cell (2023)

5. Potential Crosstalk and Baseline Fluorescence Dynamics:

To address potential crosstalk, could the authors provide raw traces showing the first 10 seconds and the last 10 seconds of recordings? Specifically, does baseline fluorescence (F) increase within the initial seconds and then saturate? This would help determine whether there is any gradual saturation effect.

Version 2:

Reviewer comments:

Reviewer #1

(Remarks to the Author)

The authors have appropriately addressed the remaining critiques.

Reviewer #3

(Remarks to the Author)

The authors have addressed the comments well. In the discussion of the prolonged IPSPs, I suggest considering the slow kinetics of the voltage sensor. This is evident in Fig 1B (black vs blue waveforms) and Supplementary Table 1.

Response to Reviewers: Carolan et al., NCOMMS-24-52289-T

We are grateful for the thoughtful and positive comments of the reviewers about our manuscript. Reviewer 1 stated: “The authors present a nice proof of principle for sub-cellular voltage measurements in response to sensory and neural stimulation, and compelling evidence for measuring stimulus-induced synaptic changes in vivo in the cerebellar cortex. The techniques are cutting-edge”. Reviewer 2 indicated that: “This is a timely work, the data is sound, and the paper is well-written.” Finally, Reviewer 3 judged that: “Overall, this work is exciting.”

To address the reviewer’s concerns we have collected new data, performed new analyses, and rewritten the manuscript. Specifically, we have:

1. Obtained new data to increase the number of cells in Fig 1. (*Reviewer 1*)
2. Clarified details regarding prevalence of optogenetically evoked EPSPs and added EPSP data from more neurons. (*Reviewer 1, Reviewer 3*)
3. Performed new analysis investigating the effect of gabazine block of IPSPs on EPSPs. (*Reviewer 1, Reviewer 3*)
4. Performed new analysis which shows brain motion is low in our preparation and that we can accurately match fields of view pre- and post-plasticity induction. (*Reviewer 1*)
5. Performed new analysis to confirm complex spike and IPSP amplitude measurements, both at the cell level and at the level of individual segments, are not sensitive to baseline fluorescence, and that our observed changes are indeed physiological. (*Reviewer 2*)
6. Performed new analysis to check for optical cross-talk, demonstrating we are not significantly activating parallel fibers during our imaging protocol. (*Reviewer 3*)
7. Performed new analysis to investigate the correlation between neurons as a function of optogenetic stimulation intensity. (*Reviewer 3*)
8. Included additional two-photon images demonstrating broad expression of ChRmine in granule cells. (*Reviewer 3*)
9. Added an entirely new set of plasticity controls including new timing configurations and testing for potentiation pre- and post-pairing. (*Reviewer 3*)
10. Included new data which shows the trial-by-trial readout of plasticity. (*Reviewer 3*)

We thank all reviewers for their thoughtful comments which have helped us to strengthen our main conclusions and craft a significantly improved manuscript.

Key

- **Blue text:** Reviewer comments
- **Regular text:** Response comments
- ***Italicised text:*** Quoted updated text
- **Underlined text:** Actions taken

Reviewer #1

The authors present a nice proof of principle for sub-cellular voltage measurements in response to sensory and neural stimulation, and compelling evidence for measuring stimulus-induced synaptic changes in vivo in the cerebellar cortex. The techniques are cutting-edge, the analysis and presentation are mostly clear, and the claims are generally appropriately matched to the data. Here are some suggestions.

We are grateful for the positive comments of this reviewer.

The elephant in the room is that the authors apparently only rarely detect EPSPs in Purkinje cells following the presumably enormous (1p) excitation of thousands of granule cells, possibly even the majority of all GrC inputs to these PCs. I can find one EPSP example (S4J) and no quantifications of the respective frequency of detection of IPSPs vs EPSPs.

1. I would appreciate some quantification of: in what fraction of cells/dendritic compartments/GrC stimulus presentations/mice the authors actually detect EPSPs and IPSPs.

The authors should then also quantify any changes in the frequency of observed EPSPs in the gabazine condition

Beyond the data analysis, at the moment, the text “just lets it slide,” which for my taste is not a great strategy. Some acknowledgement of this strange result, and some plausible explanations for what might be happening, are the minimum that would be appropriate here.

Of course I agree with the authors’ interpretation of the origin of the GrC-evoked IPSPs as reflecting feedforward inhibition through MLIs, but this does not “explain away” the dearth of preceding GrCPC EPSPs, which must come 1 synapse before the GrCMLIPC IPSP.

- Can the authors quantitatively exclude that this results from either (a) the scan rate (440 Hz) or (b) the GEVI (JEDI) being too slow to detect most EPSPs that must precede IPSPs? I.e., they are mostly getting missed/summed/blurred into a single IPSP?

- Can the authors quantitatively exclude that fast early EPSPs are missed due to the PMT gating at 1p laser pulsing (e.g., stability/kinetics of the PMT signal recovery post-gating)? The authors cite (p9LastPar) some kind of issue with PMT signals after gating, so this seems to be a problem?

- If the authors can exclude the above technical concerns, do they favor some biological explanation, e.g., globally jacking up GrCs causes Golgi cells to clamp down so there’s not much net effect (but then how to reconcile with the MLI result?).

The reviewer is correct in that we only observed clear optically induced EPSPs in a subset of recordings (n = 7 cells), corresponding to ~10% of all optogenetic stimulation experiments. In contrast 70% of our recordings showed a clear IPSP with an amplitude greater than 0.05 $\Delta F/F_0$.

We believe the primary reason for the paucity of EPSP signals is two-fold: First, physiological parallel fiber-mediated EPSPs are very brief – and therefore much smaller in size than the succeeding IPSPs – as they are almost immediately (within milliseconds) curtailed by feed-forward inhibition [Eccles JC, Ito M, Szentagothai J, The cerebellum as a neuronal machine; Springer, Heidelberg; (1967); Mittmann, W., Koch, U. & Häusser, M. *J. Physiol.* **563**, 369–378 (2005); Wulff P, Schonewille M, Renzi M, Viltono L, Sassoè-Pognetto M, Badura A, Gao Z, Hoebeek FE, van Dorp S, Wisden W, Farrant M, De Zeeuw CI, *Nature Neuroscience* 12:1042–1049 (2009); Bao J, Reim K, Sakaba T. *J Neurosci.* 30(24):8171-9 (2010); Dizon and Khodakhah, *Journal of Neuroscience* 31 (29): 10463-10473 (2011)]. Second, we have to electronically disable our PMTs while we perform widefield LED illumination, which gives rise to a short time period (5–8 ms) after stimulation where we are unable to record (with the exact time period depending on the preparation and field of view); note that this brief downtime during the optogenetic stimulus is typical across all all-optical two-photon experiments. Notably, in the cells where we observe optically induced EPSPs, the FOV exhibited minimal inhibition and fast detector recovery — the optimal conditions for EPSP detection. We have now included a new figure that summarizes the amplitude and latency of all detected EPSPs [Fig. S4K].

In addition, based on the reviewer's request, we also performed new analysis, where we assessed the effect of gabazine block of IPSPs on EPSPs. We observed that in the presence of gabazine EPSPs are broadened, which is consistent with previous findings showing that feed-forward IPSPs rapidly terminate the EPSP signal [Mittmann, W., Koch, U. & Häusser, M. *J. Physiol.* **563**, 369-378 (2005)]. We have now included this analysis in a new Supplemental Figure [Fig. S4L].

We agree with the reviewer that these issues require careful clarification and have now added the following text to the manuscript:

Main text: “Optogenetic activation of GrCs sometimes triggered all-or-none depolarizations (corresponding to parallel fiber-evoked dendritic spikes¹⁸; **Fig. S4C**), and occasionally triggered graded depolarizations (corresponding to parallel fiber EPSPs; **Fig. S4J**). These responses had smaller amplitudes than spontaneous complex spike signals ($\Delta F/F_0 = -10.5 \pm 2.9\%$, mean \pm std, $n = 7$ cells), peaked shortly after optogenetic activation (10.3 ± 0.6 ms) [**Fig. S4K**], and were shaped by inhibition [**Fig. S4L**]. This latency matches similar results in the literature¹⁹, however we note that the combination of rapid termination of EPSPs by feed-forward IPSPs^{17,20–23} and the 5-8 ms deadtime between LED onset and subsequent recording (see Methods) resulted in optically evoked EPSPs being observed only rarely.”

Methods: “Electronic gating was set to 1 ms on either side of the LED control signal, which resulted in a 5-8 ms deadtime between LED onset and subsequent voltage recording (the specific length of this period was preparation- and FOV-dependent).”

2. My other major technical concern that is basically unaddressed is brain motion. The authors cite (P9par3) a FOV 24um “tall.”

The authors should provide numbers on the peak magnitudes of X-Y motion artifacts (i.e., taken from the image registration algorithm output) during sensory and optogenetic stimulation. Even

better would be to provide these numbers from larger FOV recordings so we can be certain that the motion is fully compensated rather than being “capped” at 24um. They should thereby quantitatively demonstrate that the membranes they’re recording from aren’t simply disappearing out of the 24um FOV

For precisely the concern raised by the reviewer, we spent significant time optimizing our surgical preparation to minimize brain motion. Across all of our recordings (n = 30 for each condition) the motion, as extracted from the Suite2p registration file, under both sensory and optogenetic stimulation is shown in the below table:

	X motion (μm)	Y motion (μm)
Sensory (95 %)	0.13 ± 0.16	0.11 ± 0.11
Sensory (max)	1.60 ± 2.35	1.17 ± 1.71
Optogenetic (95 %)	0.09 ± 0.02	0.09 ± 0.03
Optogenetic (max)	1.21 ± 2.11	0.87 ± 1.53

where ‘max’ represents the maximum absolute shift in any frame throughout a single recording and ‘95 %’ represents the 95th percentile of absolute shifts across the entire recording. Quoted values are the mean ± std across n = 30 recordings. Crucially, the 95th percentile across all recordings was below a micron for each condition, demonstrating that brain motion was indeed very low throughout our experiments.

We have now added this information to the Methods section:

Methods: “Motion artifacts in our experiments were low, with 95 % of frames having motion below $X = 0.13 \pm 0.16 \mu\text{m}$ and $Y = 0.11 \pm 0.11 \mu\text{m}$ for sensory stimulation and $X = 0.09 \pm 0.02 \mu\text{m}$ and $Y = 0.09 \pm 0.03 \mu\text{m}$ for optogenetic stimulation (mean ± std across n = 30 recordings).”

They should also compare the measured motion magnitude to the expected zPSF optical section thickness to demonstrate that these very small structures aren’t vanishing from the optical section, or worse, being replaced with other small membrane segments, ideally using the density of labeled membrane segments as the relevant parameter.

The expected z-PSF is given by $2\lambda n/NA^2 = 2 \times 940 \text{ nm} \times 1.33/0.8^2 = 3.9 \mu\text{m}$, which is larger than our X and Y motion shown above.

It sounds (P11Par2) like it was not that easy to register dendrite identity 40 minutes after the “plasticity induction” protocol. Thus, it would be appropriate to provide at least examples of the first and second images of the same segments, and better yet a distribution across experiments of some image similarity metric, to demonstrate that this procedure was successful.

We have included a new Supplemental Figure [Fig. S7A,B] that shows five representative fields of view before and after our plasticity induction protocol. Visually, fields of views are well matched, however to quantify this we calculated the mean squared error (MSE) across the same fields of view before and after our plasticity induction protocol. We found $MSE = 0.016 \pm 0.009$ (mean \pm std across $n = 30$ FOVs), which was significantly smaller than a shuffle distribution of MSE between different fields of view ($MSE = 0.050 \pm 0.015$, $p = 5.66 \times 10^{-16}$ Mann-Whitney U Test) [Fig. S7C]. We are grateful to the reviewer for prompting us to perform this analysis.

A few other points

The authors cite (P9par3) “50 different fields of view from 4 mice.” I’d like to hear how the authors confirmed that they didn’t come back to the same place twice when collecting ~12 fields per animal.

To prevent collecting multiple data sets from the same field of view, we first zeroed the microscope on a common central landmark and then recorded the X,Y,Z location of each field of view to make sure we were not repeating measurements. As a further check, we also compared the mean images of each field of view. We have now added this information to the Methods section.

Fig. 1, N would ideally be more than 3.

We agree with the reviewer and have therefore collected new data sets to increase the number of cells in Figure 1 to $N = 7$.

P4Par3: “...spatial distribution...which presumably reflect” should say “reflects” or “reflected”

Fixed.

P7par3 “photomultiplier tubes set to 20” 20 what?

This refers to the gain of the particular PMT module (PMM02, Thorlabs). We have updated the Methods accordingly.

Reviewer #2

Recent advances in voltage imaging technology allow optical access to the intracellular voltage dynamics in various animal models. However, voltage imaging at subcellular resolution and particularly the optical recording of synaptic potentials are still challenging. In this work, Carolan et al demonstrate voltage imaging from superficial Purkinje cell dendrites in the cerebellum of awake mice. By inducing presynaptic activity using either optogenetics or sensory stimulation they also recorded postsynaptic potentials at dendritic resolution and managed to detect inhibitory plasticity following LTP protocol. The key technological progress enabling these nice measurements was the development of a new variant of the GEVI Jedi-2P which shows higher sensitivity at the hyperpolarized regime and thus allows high-fidelity detection of synaptic potentials. This is a timely work, the data is sound, and the paper is well-written. Below I have a few comments that I hope could further improve the manuscript.

We thank the reviewer for the positive comments.

1. Amplitudes of fluorescence events (i.e. dF/F) should be treated with care, particularly in cases of comparisons between different ROIs. Specifically, dF/F is highly sensitive to the baseline fluorescence and its amplitude is therefore quite meaningless. This particularly applies to the cross-correlation analysis between the complex spikes (FigS5I-J), and to the comparison of IPSP amplitudes between cells (Fig3C). My suggestion would be to normalize the activity of each cell to the peak of the spikes whose underlying voltage can be assumed to be equal between different cells, and then repeat all those comparisons and update the conclusions accordingly.

In principle $\Delta F/F_0$ should be insensitive to baseline fluorescence, however as the reviewer notes, in the low-SNR regime — where the fluorescence of the neuron is close to the fluorescence of the background — $\Delta F/F_0$ may be sensitive to baseline fluorescence. To test this *in vivo*, we performed a new analysis where we looked at the correlation between $\Delta F/F_0$ complex spike amplitude and baseline fluorescence across all of our data sets. We found no significant correlation ($p = 0.290$, $n = 42$ cells across four mice) [Fig. S5E].

As a further control, we also compared the mean $\Delta F/F_0$ spike amplitude at the beginning of each recording (the first 10 seconds) to the end of each recording (last 10 seconds). If we are in the low-SNR regime we may see a difference between these two data sets due to indicator photobleaching. However, across all of our data sets, we saw no significant difference in these amplitudes ($p = 0.176$ Wilcoxon-Signed rank test, $n = 43$ cells across four mice) [Fig. S5A]. We then also looked at the correlation in baseline fluorescence and IPSP amplitude, once again finding no significant correlation ($R = 0.103$, $p = 0.515$, $n = 42$ cells across four mice) [Fig. S5F]. We are grateful to the reviewer for prompting us to perform this new analysis and have included this data in an entirely new supplemental figure Fig. S5.

2. The previous point is strengthened by the data in Fig.3F which shows variability in the spike amplitude along the dendrite. I'm pretty sure that this is not physiological, and it is likely the result of variability in the indicator expression along the tree. It thus questions the conclusion that IPSP amplitude is variable. Also here, I suggest normalizing the IPSPs to the spike amplitude in each segment and then reconsidering the claims in Fig.3F-H (and also removing the two top panels in Fig.3F).

While both experimental data [Kitamura K & Häusser M. *J Neurosci.* 31(30):10847-58 (2011)]. and simulations [Zang, Y., Dieudonné, S. & De Schutter, E. *Cell Rep.* **24**, 1536–1549 (2018), Fig. 3] support the observation that complex spike signals can have variable voltage amplitudes across the dendrites of Purkinje cells, we agree with the reviewer that it is important to verify that our observed variations are physiological and not related to variations in baseline fluorescence. To test this we looked at the correlation between baseline fluorescence and $\Delta F/F_0$ complex spike amplitude across all dendritic segments. No correlation was found ($R = -0.084$ across $n = 714$ dendritic segments, $N = 43$ cells, four mice) [Fig. S5G]. Subsequently, we performed the same analysis for IPSP amplitudes, also finding no correlation between baseline fluorescence and dendritic segment IPSP amplitude ($R = 0.036$, across $n = 714$ dendritic segments and four mice). [Fig. S5H].

We have now included this analysis in a new Supplemental Figure 5. We are grateful to the reviewer for prompting us to perform this new analysis, which confirms variations in complex spike amplitude and IPSP amplitude, both across dendrites and within dendrites, are likely to be physiological, strengthening our results and highlighting one of the key advantages of our approach.

3. The exact same comment applies also for Fig.4H

This comment has been addressed by our above analysis.

Minor comment:

One advantage of this work is the usage of a conventional 2P microscope which is available in many neuroscience labs. I think that this point should be mentioned in the introduction. On the other hand, many new high-speed 2P methods have been published in recent years, and techniques such as AOD-based random access microscopy, FACED, and SMURF could significantly improve and upscale these measurements in space and time. I suggest mentioning this point in the discussion and citing the relevant works.

We have now included these points and the relevant citations in the Introduction and Discussion.

Discussion: *“Our approach is also highly compatible with recent advances in high-speed microscopy^{8,39,40}.”*

Reviewer #3

This work by Carolan and colleagues describes an optical approach by pairing one-photon bulk stimulation of granule cells with two-photon genetically targeted voltage imaging of purkinje cells to study the synaptic transmission and plasticity between granule cells and purkinje cells. For that, they develop one genetically encoded voltage indicator, JEDI-2Psub, which is expressed in purkinje cells, and use an optogenetic actuator, ChRmine, which is expressed in granule cells using a genetic mouse line (Math1-Cre). The authors are able to resolve complex spike signals and granule-cell optogenetic stimulation-induced potentials in purkinje cell dendrites.

Simultaneous optical imaging and stimulation is an exciting field, and the authors previously had multiple works on simultaneous optical Ca²⁺ imaging and optogenetic stimulation with careful characterization and discussion of the optical crosstalk. Here however, in this study, with different tools in demand of much higher power, the authors did not characterize or discuss the potential optical crosstalk (activation of the optogenetic actuator, ChRmine, when imaging JEDI-2Psub with >60mW power and with high-speed scanning).

On the biological side, the authors can resolve granule-cell optogenetic stimulation induced potentials in purkinje cell dendrites, which is impressive. However, given the long latency between the optogenetic stimulation and optically induced response, the authors should discuss why and whether this is a truly synaptic effect or multi-synaptic effect (because this will affect the interpretation of the plasticity results). Also given the potential optical crosstalk, it is unclear whether granule cells are already depolarized, or the postsynaptic dendrites maybe already depolarized or hyperpolarized.

Finally, the authors pair granule cell optogenetic stimulation with sensory stimulation for studying plasticity. Using canonical plasticity induction protocol, the authors are able to detect changes in granule-cell optical-stimulation induced voltage signals in purkinje cells. However, the authors did not address potential intrinsic excitability change during the process. As a result, the interpretation of the plasticity experiments is not convincing. Furthermore, these plasticity experiments also need further controls. And because of the multiple synapses involved, the interpretation needs to be precise in terms of whether it is plasticity of inhibitory synapses.

Overall, this work is exciting. Because of the opportunity for high interest, it is essential that the authors be held to a high standard in terms of characterization of the optical tools and interpretation of the plasticity results. Provided changes and/or new data to address the concerns, I would support the publication of the work.

We thank the reviewer for their excitement about our work and for highlighting areas where we could strengthen the manuscript. We respond to the reviewers comments and requests below.

My major points are detailed below.

1. Genetically targeted voltage imaging with JEDI-2Psub:

The ability to detect voltage in genetically identified cells in vivo is impressive. The author should acknowledge prior work of genetically targeted voltage imaging in purkinje cell dendrites (Gong, Y., ..., & Schnitzer, M. J. 2014. Nature communications).

We thank the reviewer for bringing this reference to our attention and we have now added it to the Introduction.

Secondly, the authors noted that they could detect voltage signals well in the soma. Could the author differentiate simple vs complex spikes? This needs to be discussed.

We were not able to detect simple spikes at the soma, only somatic complex spikes [Fig. S3A]. This is likely due to limitations of our imaging frame time (~2.2 ms) compared with the width of a typical simple spike, which can be under a millisecond [Llinas and Sugimori, Journal of Physiology 305: 171-195 (1980); Stuart G, Häusser M. Neuron 13(3):703-12 (1994)]. We have now clarified this in the Methods section.

Methods: *“We were also able to record somatic signals, however due to imaging framerate limitations, we were only able to record somatic complex spikes [Fig. S3].”*

Thirdly, how does the non-linearity affect the detection of subthreshold excitatory inputs to the purkinje cells? It is somewhat surprising that in fig S4H, most optogenetic stimulation induced excitatory responses are similar amplitude as complex spikes. Are they potentially big EPSPs? How many cells show clear EPSPs with a smaller amplitude compared to complex spikes as in fig S4J?

As we note in the response to Reviewer 1, we only observed optically induced EPSPs in n = 7 recordings. These signals are much smaller in amplitude ($\Delta F/F_0 = -14.2 \pm 4.0 \%$, n = 45 trials) than those associated with complex spike signals ($-31.2 \pm 4.4 \%$) and are initiated ~8 ms after the LED stimulation. In certain recordings we also see large optically induced responses [Fig. S4H]. These occur 47 ms after the optical stimulation [Fig. S4G] and have a comparable spike shape and amplitude to spontaneous complex spike signals. We have now included a new figure showing the amplitude and latency of all EPSP events [Fig. S4K].

2. Optical crosstalk between imaging and stimulation:

Does the imaging light used to image JEDI-2Psub activate the optogenetic actuator ChRmine? ChRmine is a very sensitive opsin. Neurons expressing ChRmine showed notable depolarization with high imaging power (for instance, see reference: Sridharan, Savitha, et al., Neuron. 2022, Marshel, James H., et al. Science 2019.). This crosstalk is larger in a smaller field view with high scanning speed.

This crosstalk effect is important for the interpretation of the results. For instance, would the imaging light already deplete the synaptic vesicles in granule cells? Would this already affect the synaptic inputs and resulting resting potential of the purkinje cells? Would the authors see a change in the baseline of voltage recording during the first few seconds of turning on the imaging light? Or change of spontaneous spiking rate during continuous imaging?

The focal planes of Purkinje cell dendrites and granule cell somata are different which should minimize direct granule cell activation during imaging; however we agree with the reviewer that it is important to verify we are not activating parallel fibers which may then in turn activate PCs within our imaging field of view. It has been shown *in vivo* that complex spike amplitudes are dependent on dendritic membrane potential [Kitamura, K and Häusser, M *J. Neurosci.* **31**, 10847–10858 (2011)], which is also supported by simulation studies [Zang, Y., Dieudonné, S. & De Schutter, E., *Cell Reports* 24, 1536–1549 (2018)]. If parallel fibers and in turn PCs are being activated during imaging, we therefore may expect to see a difference in complex spike amplitude across our recording. To test this, we compared the spontaneous complex spike amplitude in the first 10 s of our recording to the last 10 s of our recording. We found no significant difference [**Fig. S5A**] (Wilcoxon signed-rank test, $n = 43$ cells, $p = 0.484$). As the reviewer suggested, we also compared the spontaneous complex spike rate at the beginning of our recording to the end of our recording, again finding no significant difference ($p = 0.110$) [**Fig. S5B**].

As a further test, we also compared the complex spike amplitude between preparations with co-expression of opsin and GEVI, and GEVI-only control mice. If the imaging laser were giving rise to potentiation of PCs, we might expect to see a significant difference in spike amplitude between these two groups. However, we saw no significant difference in either complex spike amplitude [**Fig. S5C**] nor complex spike rate [**Fig. S5D**].

We have added this new analysis to a new Supplemental Figure 5, and are grateful to the reviewer for prompting us to perform this analysis which demonstrates the robustness of our approach.

3. Optogenetic recruitment of granule cells:

It is hard to see the granule cells expressing ChRmine in Fig 2C or Fig S1. Could the authors show clearer images? How many granule cells are on average recruited during optogenetic stimulation?

We have now included additional images across multiple fields-of-view and mice, showing broad expression of ChRmine in GrCs [**Fig. S1D-I**].

What is the latency between optogenetic stimulation onset to the optically induced EPSP, IPSP, and spikes?

The mean latencies, measured from LED onset to signal peak, are as follows:

- Optically induced EPSPs: 10.3 ± 0.6 ms
- Optically induced IPSPs: 85.4 ± 24.4 ms
- Optically induced spikes: 46.9 ± 16.8 ms

The reported latency in Fig. S4J is very long, presumably involving multiple synapses. Could the authors explain how they quantify the latency and why it is so long? Potentially this involves circuit effect, eg, multiple synaptic interactions among granule cells or the golgi cells or other cells. This should be discussed because it will affect the interpretation of the plasticity results in terms of which synapses are modulated.

The mean latency to the peak of our optically induced EPSPs is 10.3 ± 0.6 ms (mean \pm std, $n = 7$ cells), with a typical onset of 8 ms. This is consistent with other published data using optogenetic activation of presynaptic neurons and electrophysiological recording of the postsynaptic voltage response (e.g. Hage, T. A. *et al.*, *Elife* **11**, e71103 (2022) observe an optically induced EPSP latency of 15 ms: Figure 5, Supplemental Figure 3B). For clarity, we have now reported the amplitude and latency of all cells displaying optically induced EPSPs [Fig. S4K] and cited the relevant results from the literature.

The experiments with gabazine application are convincing that the observed hyperpolarizing events are via gabaergic synapses. With gabazine, would the authors see more optogenetically induced EPSPs and a shorter latency? If not, the authors should discuss why.

In response to this reviewer and Reviewer 1, we performed new analysis, where we looked at the effect of gabazine block of IPSPs on EPSPs. Indeed, we find that in the presence of gabazine EPSPs are broadened, which is consistent with previous findings *in vitro* showing that feed-forward IPSPs can rapidly terminate the EPSP [Mittmann, W., Koch, U. & Häusser, M. *J. Physiol.* **563**, 369–378 (2005)]. We have now included this analysis in a new Supplemental Figure [Fig. S4L].

4. Correlation analysis:

The spatiotemporal analysis is nice. Regarding the correlation of IPSP, if a lot of granule cells are recruited during the optogenetic stimulation, the neighboring purkinje cells will have high correlations of IPSPs even when they do not receive shared inhibition from presynaptic interneurons. The authors may consider analyzing the correlation under different stimulation light intensities, like in Fig 2M.

Based on the reviewer's suggestion we have now performed a new analysis that examines the change in IPSP correlation under different stimulation intensities (data from **Fig. 2M**). At 4.5 mW/mm² there is no significant correlation ($R = -0.029$, $p = 0.917$), while at 7.3 mW/mm² there is a significant correlation ($R = 0.637$, $p = 0.011$). We believe this analysis is sufficiently interesting to include it in **Fig S6K,L** and note it in the main text. An in-depth characterization of this result should be the subject of future study. We thank the reviewer for prompting us to perform this new analysis.

5. Plasticity interpretation:

The authors include controls omitting the pairing procedure. What about omitting either activation of granule cells or sensory inputs? Would the observed change be due to the animal learning of

the repeated sensory stimulation? Or could it be from the repeated optogenetic stimulation of granule cells?

While we don't have a control that omits either the activation of granule cells or sensory stimuli, we do have a control, now included in the manuscript, where we have *reversed* the timing of these two stimuli during the plasticity induction protocol, such that the airpuff occurs 80 ms *after* the LED stimulation (rather than before). If the mice were indeed learning either the repeated sensory or optogenetic stimulation we may also expect to see a significant difference in the IPSP amplitude under this condition. However, we see no significant difference (Wilcoxon signed-rank test, $p = 0.432$). We believe this control significantly strengthens our results and have therefore included this new analysis in Fig. S7B.

Secondly, could the intrinsic excitability change during the process? Does the baseline of the voltage in purkinje cells change?

If there was a change in intrinsic excitability or baseline membrane potential we might expect to see a change in spontaneous complex spike rate (reflecting a change in excitability of presynaptic neurons) or a change in complex spike amplitude (reflecting a change in postsynaptic excitability), between our pre- and post-pairing conditions. Prompted by the reviewer's question, we have performed a new analysis where we look at the difference in these two conditions, and find no significant difference in either complex spike amplitude or complex spike rate (Wilcoxon Signed-Rank test, $p = 0.239$ and $p = 0.350$ respectively). We have now included this new analysis in Fig. S7C,D and included the following text in the main text:

Main text: *"As controls, we omitted the pairing procedure and the potentiation was not observed ([Fig. 4E]; pairing group difference from the control group tested via a linear mixed-effect model with animal and FOV as random effects $p = 4.92 \times 10^{-6}$; control group $n = 32$ cells across two mice), and reversed the order of the GrC and CF stimuli which also produced no significant potentiation [Fig. S7B]. Additionally, spontaneous complex spike signals remained stable over time [Fig. S7C,D] acting as a further control to demonstrate that the measured changes in IPSP amplitude were not an artifact of photobleaching due to continued laser illumination."*

Since the observed IPSPs are at least through two synapses, one from granule cells to the inhibitory neurons, and one from the inhibitory neurons to the purkinje cells, can the author discuss where the potentiation occurs? Could it be the potentiation from granule cells to the inhibitory neurons? If not, the authors should discuss why. Otherwise, the authors should delete the claim that this is the plasticity of inhibitory synapses.

As noted above, we only observe potentiation when airpuff precedes the LED stimulation, not when the ordering is reversed. This indicates that the airpuff, and hence sensory-evoked complex spikes, are required for potentiation. As the complex spikes occur in Purkinje cells, we therefore conclude that the most likely explanation of our results is that plasticity occurs at inhibitory inputs synapsing onto Purkinje cells.

Fourthly, is there any activity change in response to sensory stimulation after the plasticity induction?

We don't have data for changes in sensory evoked complex spikes (this would require a different experimental protocol). However, as noted above, we do look at the change in *spontaneous* complex spike amplitude and rate, before and after plasticity induction, observing no significant difference [Fig. S7C,D].

Lastly, the authors mentioned long time voltage imaging multiple times. They should specify how long do they perform the voltage imaging for each session. Is it only three minutes (as indicated in the methods part)? If so, it is not 'over long time periods'.

Each imaging session lasts for three minutes and the entire imaging experiment is performed over the course of 40 minutes. We agree with the reviewer that it is important to be clear about exactly how we are making our measurements, and therefore we now use the wording '*repeatedly* over long time periods' to convey this.

Also, is the voltage imaging conducted during plasticity induction? What are the voltage dynamics during the induction process? This is important because potentially one can correlate the activity with resulted plasticity.

While we agree that capturing voltage responses during the induction protocol would be very useful, we also needed to optimise stability of the voltage signal over the experiment. To minimize any potential photobleaching which may reduce SNR and could therefore be a confound for our sensitive plasticity readout, we therefore chose not to image during the plasticity induction protocol.

6. The manuscript emphasizes 'behavior' multiple times. Could the authors quantify the behavior during optogenetic stimulation and the plasticity induction process?

While we don't have quantitative behavioural data during the plasticity induction protocol, qualitatively there were no obvious changes in mouse behaviour. Mice were habituated to the microscope and both sensory and optical stimulation multiple times before the experiment, and continued to run on the wheel during our protocol.

Minor:

1. Fig 2G: 'raw single spike' -> single complex spike

Done.

2. Fig 2J: how many trials average?

Average of 60 trials (figure caption updated).

3. Fig 4B, could the authors show the single trial IPSP data? What is the noise level like compared to the observed difference?

4. If single trial IPSPs could not show the potentiation effect, the authors should delete the claim of 'single-trial read-out of synaptic plasticity' in discussion line 2.

We are able to see potentiation at the individual trial level. We have now included this data in Fig. S7A and refer to it in the main text.

5. The authors should acknowledge prior all-optical work to probe synaptic plasticity during behavior (Deisseroth 2023 Cell, 2024 Science).

We thank the reviewer for this suggestion, and have now included these citations in the Discussion.

Response to Reviewers: Carolan et al., NCOMMS-24-52289A

We are grateful for the additional comments on our revision. These remaining concerns center on the origin of the prolonged IPSPs. To address this we have now carried out additional histological controls, performed new analyses and re-written parts of the manuscript. Specifically we have:

- Performed histology on Math1-Cre mice, revealing specific expression of opsin in granule cell (GrC) somata and parallel fibers (as well as a notable lack of expression in Purkinje cells or other brain regions). (*Reviewers 1, 3*)
- Discussed and cited previous studies which showed Purkinje cell IPSPs with similar timecourse *in vivo*. (*Reviewers 1, 3*)
- Clarified the physiological origin of the timecourse of the IPSPs. (*Reviewers 1, 3*)
- Performed new analysis to estimate the temporal synchrony between complex spike signals in the same microzone. (*Reviewer 3*)
- Shown data from the first and last 10 s of our recordings to check for saturation effects. (*Reviewer 3*)

We thank all reviewers for their thoughtful comments which have helped us to strengthen our main conclusions and craft a significantly improved manuscript.

Key

- **Blue text:** Reviewer comments
- **Regular text:** Response comments
- **Underlined text:** Actions taken

Reviewer #1 (Remarks to the Author):

Overall, the authors have addressed most of the suggestions well.

I still have a significant lingering concern about the EPSP/IPSP business that I hope the authors can address simply, as it's unlikely I will be the only one to notice this. The authors state that most optogenetically-evoked EPSPs in PCs turn on with too short a latency to see due to the ~10ms PMT gating signal recovery time—fine. They *also* state that the EPSPs are too *brief* to see even the decay from peak, and then assert that this is in turn due to the EPSP being quickly drowned by the overwhelming recruitment of interneurons—also fine. Fig. S4L shows a control/gabazine example that makes this case well, and so far the argument holds together. The problem is, in evidently almost all cases, their IPSPs are way way slower than this – with a mean optically-evoked latency of almost 100 ms (!), Fig. S6J.

We thank the reviewer for encouraging us to clarify this point further. We agree that the phrasing of 'latency' in the prior version of our manuscript was somewhat confusing. We were referring to the latency to *peak*, rather than latency to *onset*. Indeed, the potential sources of a 100 ms *onset* latency could be manifold (Reviewer 3 made a similar point). Based on the reviewers's specific feedback, we have revised and clarified our manuscript in several places and added analysis that helps to provide context for our results.

This presents multiple problems that need to be acknowledged/addressed somehow:

(1) These IPSPs cannot be the simple feedforward inhibition that curtails GrC responses—they are off by an order of magnitude. The difference in IPSP timing between Fig. S4L-gabazine (ipsp peak <20 ms) and the more typical Fig. S6J (peaks 80-100ms) makes this very clear.

We apologize for the lack of clarity in the previous version of the manuscript. We were referring to the latency to the peak of the IPSP, and have now rephrased the wording in the manuscript to refer to "time to peak". Crucially, the *onset* of the optogenetically evoked IPSPs is rapid, even though the overall timecourse (including the time-to-peak) can be prolonged. We note that *all* of our IPSPs are initiated very rapidly (9.6 ± 4.8 ms, mean time from beginning of stimulus to 10% of peak). For example, **Fig. S5H** middle panel shows an IPSP which is initiated within 7.25 ms after the beginning of the stimulus (see below). This rapid onset of the IPSPs is consistent with them being driven by feed-forward inhibition, which is typically triggered within a few milliseconds (see Mittmann et al., 2005). The time to peak and the overall duration of the resulting IPSP will depend on a range of factors, such as the firing pattern of the activated granule cells, the resulting firing pattern of the activated feed-forward interneurons, and the dynamics of the excitatory and inhibitory synapses. Even with direct stimulation of interneuron axons *in vitro* – the "ideal case" with perfectly synchronous activation of interneuron axons, which should lead to the most rapid

timecourse of the IPSP – the overall timecourse of the resulting IPSPs in Purkinje cells can be over 100 ms (e.g. see Figure 1D, Mittmann and Häusser 2007).

Fig. S5H. A representative optogenetically evoked IPSP showing the rapid IPSP onset. The same data as originally displayed as in the figure (middle panel), replotted with grid lines.

An important validation of our approach is that the time-to-peak of our optically triggered IPSPs is comparable with that of our sensory-evoked IPSPs [Fig. S7D]. This is consistent with previous work reporting that sensory stimulation can evoke IPSPs in Purkinje cells via activation of the GrC pathway triggering feedforward inhibition (see for example Supplementary Fig. 1 in Wilms & Häusser, *Nat. Comms.* (2015)), providing further evidence that our responses reflect feedforward inhibition activated by GrCs.

The IPSPs we observe in response to optogenetic activation of GrCs are also consistent with other *in vivo* results in the literature. Notably, Roome & Kuhn, *eLife* (2020) triggered IPSPs in Purkinje cells by electrical stimulation of GrCs. These IPSPs were prolonged, with a time-to-peak of ~100 ms (cf. our time-to-peak of 85.4 ms). As with our results, these IPSPs are both intensity-dependent (Roome & Kuhn, Fig. 1i,j) and blocked by GABA antagonists (Roome & Kuhn, Fig. 1j, red). The fact that they used a different method of GrC activation and obtained similar results gives further support to the conclusion that the IPSPs we observe in response to GrC stimulation reflect feedforward inhibition.

(2) Most of these IPSPs must be a bunch of synapses away from the GrC activation, perhaps even some secondary motor response to the GrC activation?

Our understanding is that the known physiology of the GrC-interneuron-Purkinje cell feedforward inhibition pathway is sufficient to explain the timecourse of our IPSPs, and that it is not necessary to invoke any long-range synaptic pathways (or “secondary motor responses”). As described above, the overall kinetics (rapid onset, prolonged timecourse) is consistent with both *in vitro* and *in vivo* work on feedforward IPSPs. There are two additional physiological parameters in the GrC-interneuron-Purkinje cell pathway which may contribute to generating more prolonged IPSPs. **First**, a single GrC stimulus can

trigger a diversity of responses in GrCs (Masoli et al., *Comm. Bio.* (2020)), with some GrCs exhibiting burst firing (Isope et al., *J Physiol.* (2003)), which in turn will prolong interneuron activation and lengthen the time course of Purkinje cell IPSPs. **Second**, optogenetic activation of GrCs in our experiments likely results in dense activation of PFs in the molecular layer, which has been shown to promote glutamate pooling and long-lasting AMPAR and NMDAR activation in molecular layer interneurons (Clark & Cull-Candy, *J. Neurosci.* (2002), Carter and Regehr *J. Neurosci.* (2000)), leading to prolonged firing of MLIs and thus providing a further mechanism for prolonging feedforward IPSPs in Purkinje cells. Indeed, the time course of the slow EPSCs observed in MLIs in response to dense PF activation (see Fig. 4 in Clark & Cull-Candy (2002)) is consistent with the IPSP time course in our experiments. **Finally**, prolonged patterns of granule cell activity (including bursting), and dense activation of parallel fibers, have both been observed *in vivo* as confirmed by electrophysiology (Rancz et al, *Nature* (2007); Powell et al., *eLife* (2015)) and imaging studies (Lanore et al., *Nat. Neurosci.* (2021); Garcia-Garcia et al., *Neuron* (2024)).

Thus, the observed timecourse of our IPSPs can be explained on the basis of the physiological features of the GrC-interneuron-Purkinje cell feedforward inhibition pathway.

(3) Again, I do not think it helps to just 'let it slide' in the text. The authors should probably disaggregate their IPSPs into a story that makes sense: (1) IPSPs that could plausibly be monosynaptic feedforward inhibition directly caused by GrC optogenetic activation (and how often were these actually seen??), and (2) then...something else, which should include some plausible physiological proposal for what the something else is, so readers don't start asking themselves the same questions I posed above.

We believe that the combination of the rapid onset, consistency with other *in vivo* results, consistency with our own sensory-evoked IPSPs and the physiological explanation we provide above (point 2), gives confidence that our evoked IPSPs reflect feedforward inhibition from GrCs. We agree with the reviewer that it is important to clarify their origin and we have now:

- Explicitly noted the IPSP onset time and peak latency in the main text.
- Referenced other *in vivo* results which demonstrate comparable IPSP time courses (Roome & Kuhn, *eLife* (2020)).
- Provided a likely physiological explanation for the duration of our IPSP responses in the main text.

(4) this also assumes (hopefully) that the authors actually have evidence for the monosynaptic feedforward inhibition, since such fast IPSPs are also the authors' proposed explanation for the absence of EPSPs in the data (that they get shut down within <20 ms).

As we discuss above, while the IPSPs we observe have a slow time course, they have a fast onset (within a few milliseconds), which explains the absence of EPSPs in our data.

(5) Is there another explanation for all of this? I started looking in more detail at how the authors did the optogenetics, and I see that they used ChRmine virus in adult Math1-Cre mice. No histology is shown. I apologize that I did not comment on this in the first round, but it is important to nail down. To my knowledge, *Math1-Cre has no cre expression in adult cerebellar granule cells* (10.1016/j.neuron.2005.08.028)—it expresses in progenitors and thus is only suited for granule cell expression when used in *crosses*. This suggests that whatever neurons the authors have successfully infected here is somehow due to cre-independent leak expression. This then raises the question of what else is expressing ChRmine. I am not normally a stickler for providing histology, but since the authors' strategy should in principle not have worked at all, it seems to be very necessary. The authors should ideally provide histology that includes the cerebellum and some major pre-cerebellar nuclei like the pons. My point in bringing this up is in trying to understand the confusing above results (lack of EPSPs and also possibly lack of short-latency monosynaptic feedforward inhibition IPSPs). One possibility is that the authors are not only (or not primarily) activating granule cells.

To achieve expression of ChRmine specifically in granule cells, we injected adult Math1-Cre transgenic mice (line B6.Cg-Tg(Atoh1-cre)1Bfri/J, JAX #011104) with Cre-dependent ChRmine virus. While this line uses the promoter elements of Math1 and is therefore useful for specific expression of constructs in cerebellar GrCs, it is a transgenic line and therefore expression of the opsin is not subjected to the same regulatory control as the endogenous Math1 gene, which, as the reviewer points out, is not expressed in adult animals [R. Machold & G. Fishell, *Neuron* (2005)]. A detailed discussion of the possible reasons for these types of expression differences has been provided in the literature (Laboulaye et al., *Front. Mol. Neurosci.* (2018), Luo et al., *Neuron*, (2020)).

That said, we agree with the reviewer that it is important to verify the specificity of our expression strategy in GrC. We have therefore included new experiments in which we injected a Cre-dependent ChR2-TdTomato fusion protein into our Math1-Cre mice and performed histology to assess the specificity of opsin expression [Fig. S2]. These experiments reveal expression of fluorescence in GrC bodies and parallel fibers (as well as a notable lack of expression in Purkinje cells or in other brain regions). This analysis confirms that our expression strategy achieves specific opsin expression in GrCs. We are grateful to the reviewer for prompting us to include these new experiments, which we believe has significantly strengthened our manuscript.

Reviewer #2 (Remarks to the Author):

The authors have addressed my comments well and I don't have further questions.

We are delighted that we have addressed all of the reviewers comments and are grateful for their help in strengthening our manuscript.

Reviewer #3 (Remarks to the Author):

The authors have made an effort to enhance the manuscript. The primary biological claim is that pairing optogenetic stimulation of granule cells with sensory stimulation can trigger long-term potentiation of inhibitory synapses. However, without further analysis and appropriate controls, this claim should be tempered throughout the paper.

Major:

1. Latency of IPSP Response:

The long latency between optogenetic stimulation and IPSP (85.4 ± 24.4 ms) should be explicitly stated in the main text, along with a discussion of its implications. Feed-forward inhibition (mentioned on lines 113 and 122) typically occurs within 1–2 ms, as cited in Mittmann et al. (2005):

"We have demonstrated directly that FFI is activated by PF input in just over 1 ms on average, with delays of less than 1 ms observed in some neurons."

Given this, how can feed-forward inhibition explain the observed ~85 ms latency? This discrepancy needs further clarification.

We apologize for the confusion. We used the term 'latency' to refer to the time of the IPSP *peak* following the optical stimulation, not the time of IPSP *onset*. We have clarified this in the text and have also provided a detailed explanation of what likely underlies the IPSP timecourse in our response to Reviewer #1. In short, though the mean peak latency of the IPSPs is ~85 ms, the IPSPs have a fast onset, typically within a few milliseconds. Our IPSP peak latencies are consistent with other published *in vivo* results (e.g. Roome & Kuhn, *eLife* 2020) as well as our own sensory activated IPSPs, and can be explained by the physiological features of the granule cell-interneuron-Purkinje cell feedforward inhibition pathway, including diversity of granule cell responses to a single stimulus, and dense activation of parallel fibers leading to long-lasting AMPAR and NMDAR activation in molecular layer interneurons (Clark & Cull-Candy, *J. Neurosci.* (2002)).

We have now clarified that IPSP onset is rapid and (to avoid confusion) rephrased the IPSP peak latency as time-to-peak in the main text. We have also cited the relevant *in vivo* literature, which is consistent with our results, and provided a likely physiological explanation for the duration of our IPSPs. In addition, we have performed new histological controls which demonstrate the specificity of opsin expression in GrCs and parallel fibers with our approach.

2. Noise Level in dF/F Measurements:

The noise level should be clearly specified in dF/F format in both the main text and Figures 2 and 4. From a visual assessment of Fig. 2E and F, and the d' value (line 93), the noise level appears to be around 3–5% dF/F. If this estimation is accurate, how is it possible to detect the single-trial plasticity depicted in Fig. 4, where the average increase is only ~2% dF/F—lower than the mean noise level? The authors should address this concern explicitly.

When analysing data to detect changes in IPSP amplitude in response to induction protocols, we averaged multiple trials ($n = 60$), enabling readout of amplitude changes smaller than our trial-to-trial variability (e.g. **Fig. 4C**). To avoid ambiguity we have now removed the phrase ‘single-trial readout of synaptic plasticity’. At the reviewer’s request, we have also included explicit $\Delta F/F_0$ values with errors in the main text and in the captions of Figs. 2 and 4.

3. Imaging Rate and Temporal Resolution:

In line 148, please specify the imaging rate. If the imaging rate is 440 Hz, how can it reliably detect temporal synchrony of 1.93 ms—less than the duration of a single frame? This should be explained in more detail.

We re-analysed our data using an established method (Stark and Abeles, *J. Neurosci. Methods* 2009), finding that temporal synchrony is indeed greater than the duration of a single frame. The results of this new analysis do not affect our overall conclusions. We have now updated the text in the results section of our manuscript and added a new section to our Methods section explaining our analysis. We are grateful to the reviewer for prompting us to reconsider this analysis.

4. Context of Voltage Imaging Advances:

While the study effectively demonstrates voltage imaging at subcellular resolution, recent advances using one-photon voltage imaging have also achieved subcellular resolution and enabled all-optical interrogation of synaptic plasticity when combined with optogenetics. These works should be at least acknowledged: Park et al., *Nature Communications* (2025), Wong-Campos et al., *bioRxiv* (2023), Deisseroth, *Cell* (2023)

We are grateful to the reviewer for the suggestion. We have now cited all of these studies.

5. Potential Crosstalk and Baseline Fluorescence Dynamics:

To address potential crosstalk, could the authors provide raw traces showing the first 10 seconds and the last 10 seconds of recordings? Specifically, does baseline fluorescence (F) increase within the initial seconds and then saturate? This would help determine whether there is any gradual saturation effect.

We have now added a new figure where we show raw traces from the first 10 s and last 10 s of a three minute recording (Fig. S1N,O). We see no evidence of an increase in fluorescence in the first few seconds of our recordings.

Response to Referees, NCOMMS-24-52289B (Carolan et al.)

Reviewer #1 (Remarks to the Author):

The authors have appropriately addressed the remaining critiques.

We are pleased that we have satisfactorily addressed all of the comments of this reviewer.

Reviewer #3 (Remarks to the Author):

The authors have addressed the comments well. In the discussion of the prolonged IPSPs, I suggest considering the slow kinetics of the voltage sensor. This is evident in Fig 1B (black vs blue waveforms) and Supplementary Table 1.

We appreciate the suggestion and have now included the possibility that the slow kinetics of the voltage sensor may contribute to the timecourse of the prolonged IPSPs in the manuscript:

“Their time course is likely shaped, in part, by the dynamics of GrC spiking²³, by the prolonged activation of interneurons by dense parallel fiber activation²⁴ (as has been observed in vivo under physiological conditions^{25,26}) and by the kinetics of the GEVI.”

This explicitly proposes that the kinetics of the voltage sensor may contribute to the prolonged IPSP, as suggested by the referee.